# The ion channel CALHM6 controls bacterial infection-induced cellular cross-talk at the immunological synapse

Sara Danielli[1,†], Zhongming Ma[2,†] (iD), Eirini Pantazi[1], Amrendra Kumar[3,4], Benjamin Demarco[1], Fabian A Fischer[1] (iD), Usha Paudel[2], Jillian Weissenrieder[2] (iD), Robert J Lee[2,5], Sebastian Joyce[3,4], J Kevin Foskett[2,6,*] (iD) & Jelena S Bezbradica[1,**] (iD)

## Abstract

Membrane ion channels of the calcium homeostasis modulator (CALHM) family promote cell–cell crosstalk at neuronal synapses via ATP release, where ATP acts as a neurotransmitter. CALHM6, the only CALHM highly expressed in immune cells, has been linked to the induction of natural killer (NK) cell anti-tumour activity. However, its mechanism of action and broader functions in the immune system remain unclear. Here, we generated *Calhm6*$^{-/-}$ mice and report that CALHM6 is important for the regulation of the early innate control of *Listeria monocytogenes* infection *in vivo*. We find that CALHM6 is upregulated in macrophages by pathogen-derived signals and that it relocates from the intracellular compartment to the macrophage-NK cell synapse, facilitating ATP release and controlling the kinetics of NK cell activation. Anti-inflammatory cytokines terminate CALHM6 expression. CALHM6 forms an ion channel when expressed in the plasma membrane of *Xenopus* oocytes, where channel opening is controlled by a conserved acidic residue, E119. In mammalian cells, CALHM6 is localised to intracellular compartments. Our results contribute to the understanding of neurotransmitter-like signal exchange between immune cells that fine-tunes the timing of innate immune responses.

**Keywords** CALHM; immune synapse; infection; macrophages; NK cells
**Subject Categories** Immunology; Microbiology, Virology & Host Pathogen Interaction
**The EMBO Journal (2023) 42: e111450**

## Introduction

Myeloid cells, such as macrophages and dendritic cells (DC) control the type, magnitude and duration of immune responses to infection. Macrophages and DCs instruct the functions of both innate and adaptive effector cells, including natural killer (NK) and T cells. NK cells can be activated directly upon recognition of infected or transformed cells, or indirectly upon priming by activated macrophages and DCs (Newman & Riley, 2007). The latter potently activates NK cells *in vivo* in a process that requires the formation of an immunological synapse (IS) between activated macrophages or DCs, and NK cells. At the IS, myeloid cells and NK cells exchange soluble and contact-dependent activating signals (Newman & Riley, 2007). Once activated, NK cells deploy cytolytic activity to eliminate infected and malignantly transformed cells and contribute to the regulation of adaptive cells by producing cytokines including IFN-γ, IL-10 and TNF (Vivier *et al*, 2008).

Ebihara *et al* (2010) identified calcium homeostasis modulator 6 (CALHM6), also known as IFN regulatory factor 3-dependent NK-activating molecule (INAM), as a membrane protein induced in activated macrophages and DCs. CALHM6 was required to induce IFN-γ secretion, but not cytotoxicity, from NK cells. Consequently, CALHM6 deficiency resulted in poor anti-tumour immunity against IFN-sensitive metastatic tumours and reduced efficacy of IFN-dependent, poly(I:C)-based immunotherapy (Kasamatsu *et al*, 2014). These findings raise several questions: how CALHM6 expression is regulated in myeloid cells during inflammation, how CALHM6 contributes to the activation of NK cells, and whether CALHM6 controls NK cell responses during infection.

Members of the CALHM family are transmembrane proteins, some of which form voltage-gated oligomeric large-pore ion- and

1 The Kennedy Institute of Rheumatology, University of Oxford, Oxford, UK
2 Department of Physiology, Perelman School of Medicine, University of Pennsylvania, Philadelphia, PA, USA
3 Department of Veterans Affairs, Tennessee Valley Healthcare System, Nashville, TN, USA
4 Department of Pathology, Microbiology, & Immunology, Vanderbilt University Medical Center, Nashville, TN, USA
5 Department of Otorhinolaryngology, Perelman School of Medicine, University of Pennsylvania, Philadelphia, PA, USA
6 Department of Cell and Developmental Biology, Perelman School of Medicine, University of Pennsylvania, Philadelphia, PA, USA
  *Corresponding author. Tel: +1 215 898 1354; E-mail: foskett@pennmedicine.upenn.edu
  **Corresponding author. Tel: +44 1865 612 622; E-mail: jelena.bezbradica@kennedy.ox.ac.uk
  †These authors contributed equally to this work

adenosine triphosphate (ATP)-permeable channels (Ma *et al*, 2016; Taruno, 2018). All CALHM monomers have four transmembrane domains, with cytosolically positioned C and N termini (NT) (Siebert *et al*, 2013). Like pannexins and connexins (other families of better-studied large-pore channels), CALHM monomers oligomerise to form channels. CALHM channels are made of eight (Ma *et al*, 2012, 2018), 10 (Drozdzyk *et al*, 2020; Liu *et al*, 2020), 11 (Choi *et al*, 2019; Drozdzyk *et al*, 2020) or 13 monomers (Liu *et al*, 2020), and can be permeable to ATP, Ca$^{2+}$, and monovalent ions (Ma *et al*, 2012; Romanov *et al*, 2018). The best-studied physiological role of CALHMs is in taste perception (Romanov *et al*, 2018). In taste-bud cells, CALHM1 and CALHM3 hetero-oligomerise to form a voltage-gated ion channel located in the synapse between a taste-bud cell and a sensory neuron. Sweet, bitter, salt or umami substances trigger Na$^+$ action potentials that activate CALHM1/3 channels, promoting the release of ATP as a neurotransmitter for gustatory neurons (Romanov *et al*, 2018). Mice lacking CALHM1 or CALHM3 lose the ability to taste sweet, umami and bitter flavours (Taruno *et al*, 2013). Beyond the neuronal system, extracellular ATP has signalling functions in the immune system as well. Within damaged tissues, high levels of extracellular ATP released by dying cells act as a danger signal (Junger, 2011) to activate the inflammasome in myeloid cells (Gombault *et al*, 2012). ATP can also act locally in the IS as a neurotransmitter-like signalling molecule. ATP released into the IS upon T cell receptor engagement activates P2X receptors in an autocrine manner, acting as a co-stimulator for T cell activation (Schenk *et al*, 2008). CALHM6 (gene name FAM26F, for "family with sequence similarity 26, member F") (Ebihara *et al*, 2010), is the only member of the CALHM family that is highly expressed in immune cells, especially in myeloid cells, but whether it localises to the IS to act as an immune synapse ATP-permeable ion channel remains unknown.

To address the role of CALHM6 in the immune system during infection, we focused on the intracellular pathogen *Listeria monocytogenes* (*L. monocytogenes*), a Gram-positive bacterium that can cause serious infections in children, elderly, immunocompromised and pregnant people (Radoshevich & Cossart, 2018). Early control of *L. monocytogenes* before adaptive T and B cell responses are initiated depends on NK cells, IFN-γ and myeloid cells, the three components of the immune system previously linked to CALHM6 biology in tumour studies. During the first few days of *L. monocytogenes* infection, IFN-γ is strongly induced, primarily by NK cells (Kubota & Kadoya, 2011) activated by macrophages and DCs after they have detected the presence of *L. monocytogenes*. NK cell-derived IFN-γ boosts the antimicrobial programme of macrophages and DCs, enabling them to kill *L. monocytogenes* and establish early innate control of infection (Portnoy *et al*, 1989; Shaughnessy & Swanson, 2007). Mice incapable of either producing IFN-γ or responding to it are highly susceptible to *L. monocytogenes* (Harty & Bevan, 1995; Roesler *et al*, 1999). Hence, we hypothesised that CALHM6 is required for myeloid instruction of NK cell activation upon infection and that the deletion of CALHM6 would impair early innate control of *L. monocytogenes*. Accordingly, we also set out to characterise the mechanism by which CALHM6 facilitates communication between innate cells.

# Results

## CALHM6 expression in myeloid cells is upregulated by pro-inflammatory and downregulated by anti-inflammatory signals

CALHM6 is the only member of its protein family highly expressed in immune cells. The Immunological Genome Project database (Heng *et al*, 2008), shows *Calhm6* mRNA preferentially expressed by myeloid cells in humans and mice, with steady-state expression detected only in mouse splenic red pulp macrophages and blood Ly6c$^+$ monocytes. Stimuli such as type I IFN and Poly(I:C) have been linked to *Calhm6* upregulation (Ebihara *et al*, 2010; Kasamatsu *et al*, 2014). To identify the myeloid cells with the highest upregulation of *Calhm6* upon stimulation *in vivo*, we injected mice with Poly (I:C). After 6 h we isolated splenic macrophages (defined here as F4/80$^+$ cells) and DCs (CD11c$^+$ cells) and found that *Calhm6* mRNA was much more strongly upregulated in splenic macrophages (Fig 1A). These results were recapitulated when we measured CALHM6 protein expression in stimulated bone-marrow derived macrophages (BMDM) and bone marrow-derived dendritic cells (BMDC). CALHM6 was absent at steady state but became highly upregulated in BMDM, and less so in BMDC, when stimulated with Poly(I:C), IFN-γ or a combination of these two signals *in vitro* (Fig 1B). In Western blots, some CALHM6 was present as a 37 kDa protein, expected for the monomer, but the majority was observed as a fuzzy band around 50 kDa, previously identified as an N-glycosylated form (Ebihara *et al*, 2010). Based on our gene and protein expression data, and unlike previous studies that focused on DCs (Ebihara *et al*, 2010), we decided to investigate CALHM6 biology in macrophages as the cell type with the highest CALHM6 expression. Our focus on macrophages in this study, however, does not exclude the possibility that CALHM6 is important in DCs in other conditions (e.g. upon detection of tumours, where its expression may be induced at higher levels; Chiba *et al*, 2014).

To define signals that regulate the expression of CALHM6, we measured mRNA expression of all *Calhm* family members, and protein expression of CALHM6, in BMDM from WT mice. Cells were stimulated for 6 or 24 h with pathogen-derived signals including LPS, Poly(I:C), Zymosan or with the pro-inflammatory cytokine IFN-γ (Fig 1C and D). None of the CALHMs were expressed in non-stimulated conditions, whereas CALHM6 was the only family member whose mRNA expression was upregulated in BMDM by pro-inflammatory signals (Fig 1C), with upregulated protein detected already within 2 h of stimulation (Fig 1D). Both LPS and IFN-γ upregulated CALHM6, with high and sustained levels of CALHM6 expression induced by IFN-γ, while LPS-induced expression was lower and transient over 48 h (Fig 1D). As LPS induces a strong IL-10-mediated negative feedback loop to limit excessive inflammation (Iyer *et al*, 2010), we next tested how CALHM6 responds to stimuli generally defined as anti-inflammatory (TGF-β, IL-4 and IL-10) (Fig 1E and F). All the anti-inflammatory stimuli tested downregulated the expression of *Calhm6* mRNA (Fig 1E) and CALHM6 protein (Fig 1F), independent of LPS or IFN-γ as the inductive signals. None of the anti-inflammatory signals induced the expression of other members of the CALHM family (Fig 1E).

Macrophage responses to inflammatory stimuli are often dependent on the integration of signals emanating from cytokine receptors

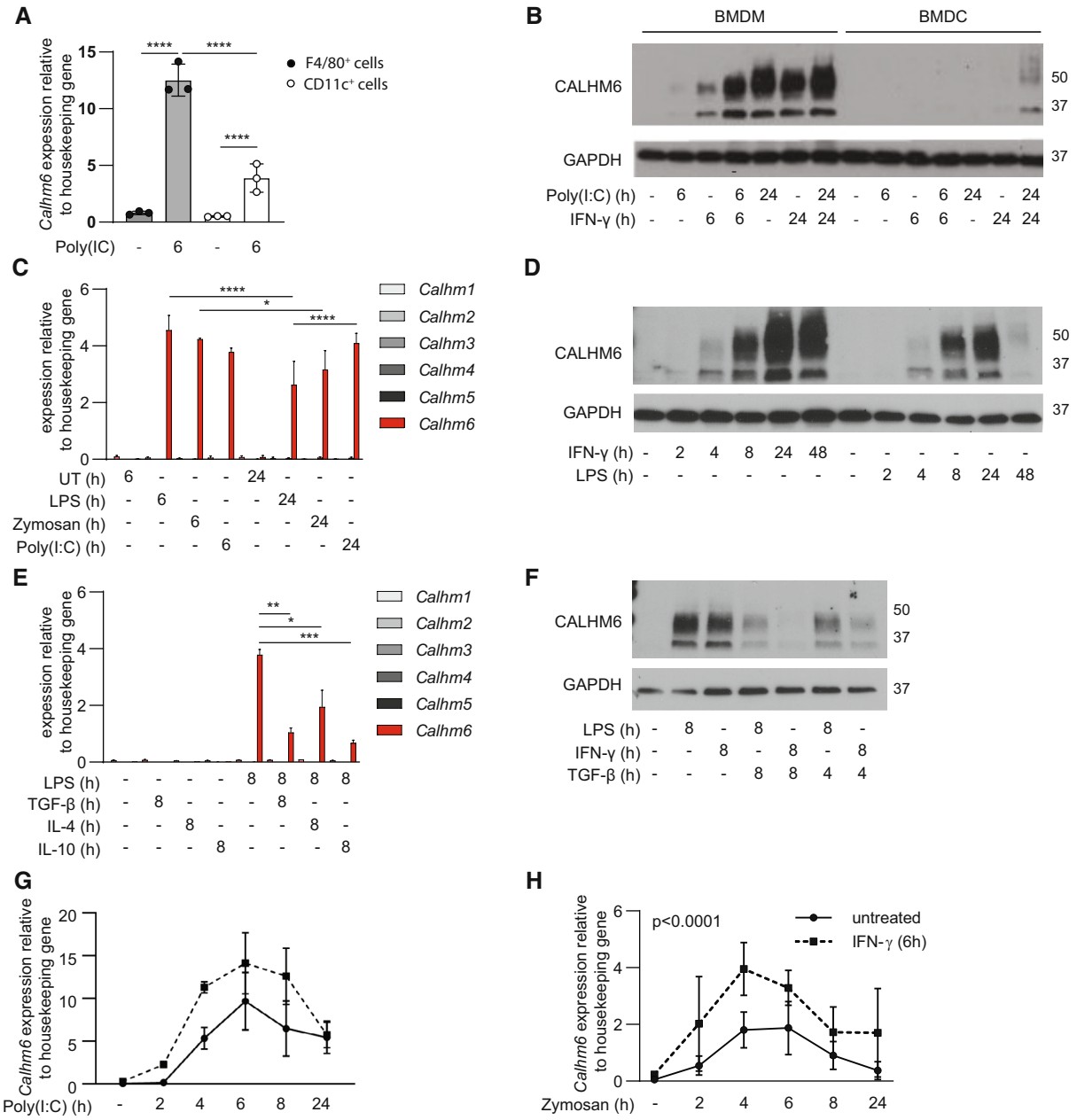

**Figure 1. CALHM6 is upregulated by pro-inflammatory and downregulated by anti-inflammatory signals.**

A *Calhm6* mRNA expression measured by qPCR in F4/80[+] macrophages and CD11c[+] DCs magnetically isolated from spleens of mice injected with Poly(I:C) (0.5 mg/mouse) or PBS for 6 h (Poly(I:C)-treated mice *n* = 3, control mice *n* = 3).

B Representative immunoblot analysis of CALHM6 expression in BMDM and BMDC stimulated for 6 or 24 h with Poly(I:C) (50 µg/ml) or LPS (100 ng/ml) (one representative experiment of two is shown, *n* = 2 mice).

C BMDM were stimulated for 6 or 24 h with LPS (100 ng/ml), Zymosan (100 µg/ml) or Poly(I:C) (50 µg/ml). *Calhms* mRNA expression measured by qPCR (one representative experiment of two is shown, *n* = 3 mice per experiment, total *n* = 6 mice, biological replicates).

D Representative immunoblot analysis of CALHM6 expression in BMDM stimulated with LPS (100 ng/ml) or IFN-γ (10 ng/ml) for 2–48 h (one representative experiment of three is shown, *n* = 3 mice).

E *Calhms* mRNA expression measured by qPCR in BMDM stimulated for 8 h with LPS (100 ng/ml), alone or together with recombinant TGF-β (10 ng/ml), IL-10 (10 ng/ml) or IL-4 (10 ng/ml) (one representative experiment of two is shown, *n* = 9 mice, biological replicates).

F Representative immunoblot analysis of CALHM6 expression in BMDM stimulated with LPS (100 ng/ml) or IFN-γ (10 ng/ml) with or without TGF-β (10 ng/ml) for 8 or 4 h (one representative experiment of three is shown, *n* = 3 mice).

G, H *Calhm6* mRNA expression measured by qPCR in naïve or IFN-γ (10 ng/ml) primed (6 h) BMDM stimulated with (G) Poly(I:C) (50 µg/ml) (one representative experiment of two is shown, *n* = 2 mice, biological replicates), or (H) Zymosan (100 µg/ml) for 2–24 h (one representative experiment is shown, *n* = 3 mice, biological replicates).

Data information: *P < 0.05, **P < 0.01, ***P < 0.001, ****P < 0.0001 Statistical test: Two-way ANOVA, error bars represent SD.
Source data are available online for this figure.

and pattern recognition receptors (PRR) (Bezbradica & Medzhitov, 2009; Bezbradica *et al*, 2014). The *Calhm6* gene promoter contains STAT1- and interferon-stimulated response element binding sites (Malik *et al*, 2017). Hence, we tested if IFN-γ boosts *Calhm6* expression even higher in the presence of pathogen-derived signals. To mimic cytokine-mediated priming of BMDM, IFN-γ was administered for 6 h before Poly(I:C) (Fig 1G) or Zymosan (Fig 1H). In both cases, IFN-γ priming significantly increased the magnitude of *Calhm6* mRNA expression. In summary, CALHM6 expression is finely regulated: macrophages primed with pro-inflammatory cytokines express high levels of CALHM6 when subsequently exposed to pathogen-derived signals. Conversely, in the presence of anti-inflammatory signals, CALHM6 expression is terminated. We conclude that CALHM6 behaves like an early pro-inflammatory protein in macrophages.

## *Calhm6*$^{-/-}$ mice show normal immune development at steady state

To elucidate the biology of CALHM6 and its role in infection *in vivo*, we generated conditional CALHM6-deficient mice carrying a LacZ reporter allele, using embryonic stem cells from EUCOMM repository (Friedel *et al*, 2007) (Fig EV1A). To generate a germline *Calhm6* deficient mouse line, we crossed C57BL/6N *Calhm6*$^{fl/fl}$ mice with the C57BL/6J PGKCre line to delete the floxed CALHM6 embryonically in all tissues (Fig EV1B). Cre-mediated deletion resulted in the removal of the C-terminal cytoplasmic region and transmembrane domains of CALHM6 (codons 198–364). The resulting *Calhm6*$^{-/-}$ mouse is not dissimilar from the one developed by Kasamatsu *et al* (2014) but was generated independently. WT and *Calhm6*$^{-/-}$ mice were born at the expected Mendelian ratio and showed no visible abnormalities when maintained under specific pathogen-free conditions. Immune-cell composition of the spleen and thymus of adult mice did not show any differences in either adaptive (Fig EV1C and E) or innate (Fig EV1D) immune cell abundance. These data are consistent with the finding that CALHM6 is expressed at very low levels in the absence of inflammatory signals (Fig 1). We conclude that CALHM6 is dispensable for healthy development of C57BL/6 mice and for the maturation of their immune system.

## CALHM6 is important for the early innate control of *L. monocytogenes* infection

The innate immune response plays a critical role in controlling the early stages of *L. monocytogenes* infection (Pamer, 2004). After intraperitoneal (i.p.) or intravenous injection, *L. monocytogenes* burden in infected mice peaks in the spleen and liver around day 3 post-injection, and bacteria are normally cleared by day 7 when the adaptive response becomes activated (D'Orazio, 2019). As innate control of *L. monocytogenes* depends on myeloid cells, NK cells and IFN-γ, the three components of the immune system previously linked to CALHM6 biology in tumour studies, we tested whether CALHM6 deficiency affected early control of *L. monocytogenes* infection. We injected (i.p.) WT and *Calhm6*$^{-/-}$ mice with $3 \times 10^5$ colony forming units (CFU) of *L. monocytogenes* and measured bacterial burdens in the spleen 1, 3 and 7 days post-injection (Fig 2A), as well as weight changes up to peak infection at day 3. Since all mice received the same dose of the pathogen, there was no

difference in splenic bacterial load after 24 h, as expected (Fig 2B). In contrast, 72 h later, at the typical peak of infection, *Calhm6*$^{-/-}$ mice had a significantly higher bacterial burden in the spleen (Fig 2C) and they were generally more affected by the infection, as shown by a significantly higher weight loss (Fig 2D). By day 7 post-infection, when the adaptive response is typically generated (D'Orazio, 2019), both WT and *Calhm6*$^{-/-}$ achieved sterilising immunity (Fig 2E). In line with the higher bacterial load at peak infection, on day 3 *Calhm6*$^{-/-}$ mice showed a higher inflammatory response in serum, with increased levels of pro-inflammatory cytokines IFN-γ and IL-6, and lower levels of anti-inflammatory IL-10 (Fig 2F–H). Similar results were reproduced when mice were infected with 10 times lower levels of *L. monocytogenes* (Fig EV1F and G). Collectively these findings demonstrate that *Calhm6*$^{-/-}$ mice have impaired early innate control of *L. monocytogenes* at the peak of infection *in vivo* but retain intact late control that allows them to clear bacteria after 7 days.

To test whether adaptive responses are indeed intact in *Calhm6*$^{-/-}$ mice, we injected WT and *Calhm6*$^{-/-}$ mice with a non-lethal dose of *L. monocytogenes*, then left to recover for 6 weeks, enough time to form adaptive immune memory. After 42 days post-injection, mice were challenged with 10 times more *L. monocytogenes* ($3 \times 10^6$ CFU) (Fig EV1H). Both WT and *Calhm6*$^{-/-}$ mice had a similar bacterial burden in spleen and liver 3 days post-challenge (Fig EV1E and J), suggesting that *Calhm6*$^{-/-}$ mice have no deficits in developing long-term memory and adaptive immunity.

## CALHM6 controls the kinetics of IFN-γ production by NK cells

Efficient innate control of *L. monocytogenes* infection relies on early NK cell activation within the first 24 h post-infection. NK cells are the main early producers of IFN-γ, the cytokine required to boost antimicrobial responses in macrophages and to establish innate control of infection (Portnoy *et al*, 1989; Nomura *et al*, 2002; Pamer, 2004). CALHM6 was previously shown to play a role in NK cell activation upon recognition of tumours, and Poly(I:C) exposure (Chiba *et al*, 2014; Kasamatsu *et al*, 2014). Because we showed that *Calhm6*$^{-/-}$ mice had poor innate control over bacterial load at peak infection, day 3 (Fig 2), we hypothesised that *Calhm6*$^{-/-}$ mice have very early defects in NK cell activation and IFN-γ secretion, which results in the observed delay in innate responses resulting in poor control of infection. We also predicted that NK cells eventually become activated, albeit with delayed kinetics which allows *Calhm6*$^{-/-}$ mice to launch the inflammatory response we detected by day 3, and to clear bacteria by day 7 (Fig 2). To test the "delayed kinetics" hypothesis, we injected WT and *Calhm6*$^{-/-}$ mice with *L. monocytogenes* i.p. and analysed NK activation, either early, 18 h post-infection when IFN-γ production from NK cells is at its peak in WT mice (Kubota & Kadoya, 2011), or late, 3 days post-infection, to detect delayed NK cell activity. To visualise cytokine production in flow cytometry-gated NK cells, splenocytes from infected mice were cultured for an additional 4 h *ex vivo* with Brefeldin A, to block ER transport and protein secretion (Doms *et al*, 1989) and accumulate any IFN-γ inside the NK cells. In support of a "delayed kinetics" hypothesis, *Calhm6*$^{-/-}$ NK cells barely produced IFN-γ at the early stage, that is 18 h post-infection, when compared to infected WT controls (Fig 3A and B). At this early time, NK cells were the main producers of IFN-γ in WT mice, and hence were the cells most

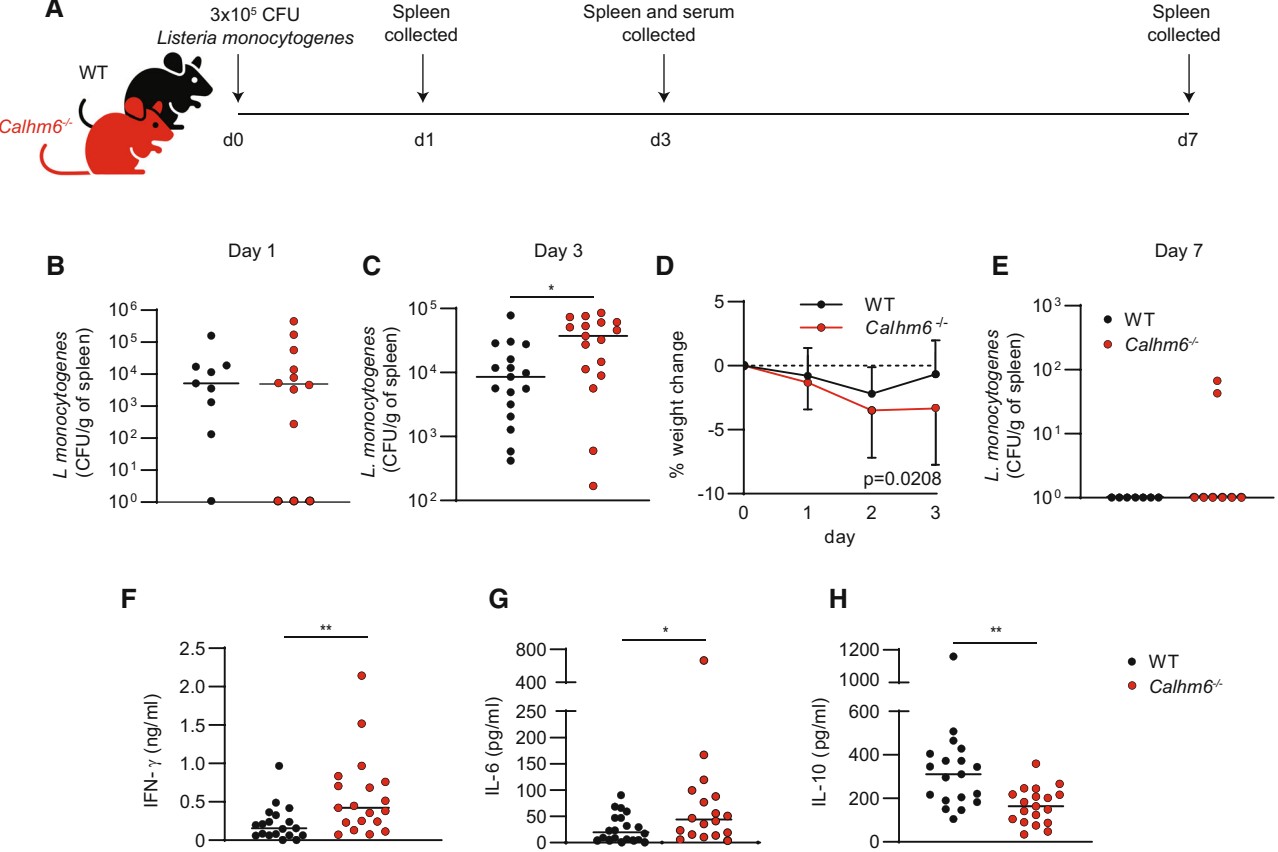

**Figure 2.** *Calhm6*$^{-/-}$ mice poorly control *Listeria monocytogenes* burden at the peak of infection.

A    Experimental scheme: WT and *Calhm6*$^{-/-}$ mice were injected i.p. with 3 × 10$^5$ CFU of *L. monocytogenes*. On day 1, 3 and 7, mice were sacrificed, spleens mechanically dispersed using beads, and cells grown on antibiotic-free BHI agar plates to determine CFUs.

B–H  CFU/g of spleen are shown (B) after 24 h (pooled results from two independent experiments, WT mice = 9, *Calhm6*$^{-/-}$ mice = 12, Man Whitney test), (C) 72 h (pooled results from two independent experiments, WT mice = 16, *Calhm6*$^{-/-}$ mice = 17, Man Whitney test) and (E) 7 days (results from one experiment, WT mice = 7, *Calhm6*$^{-/-}$ mice = 8, Man Whitney test). (D) Weight loss monitored daily for 3 days post-inoculation (pooled results from two independent experiments, WT mice = 16, *Calhm6*$^{-/-}$ mice = 17, two-way ANOVA). Serum collected from infected mice at peak infection (day 3) and serum cytokines quantified using LEGEN-Dplex Mouse Inflammation Panel assay to determine IFN-γ (F), IL-6 (G) and IL-10 (H) concentration (pooled results from two independent experiments, WT mice = 16, *Calhm6*$^{-/-}$ mice = 17, Mann–Whitney test).

Data information: *$P < 0.05$, **$P < 0.01$.
Source data are available online for this figure.

affected by CALHM6 deficiency (Fig 3C). As we further predicted, by day 3 post *L. monocytogenes* injection, *Calhm6*$^{-/-}$ NK cells became activated with delayed kinetics and produced similar, if not higher, levels of IFN-γ when compared to WT controls (Fig 3D and E). The "delayed kinetics" of NK cell activation in *Calhm6*$^{-/-}$ mice was not restricted to the *L. monocytogenes* infection model, as we found a similar phenotype after systemic innate stimulation with Poly(I:C) *in vivo*. In this model too, early activation of NK cells, that is 3 h post Poly(I:C), was impaired in *Calhm6*$^{-/-}$ mice (Fig 3F and G), whereas they became activated with delayed kinetics at later times, that is 12 h post Poly(I:C) injection (Fig 3I and J). In the Poly(I:C) model, the main early producers of IFN-γ in WT mice are also NK cells, and hence were the cells most affected by CALHM6 deficiency *in vivo* (Fig 3H). After Poly(I:C) stimulation, there are two distinct subsets of early activated NK cells, one "cytotoxic" that upregulates Granzyme B and another "pro-inflammatory" that produces IFN-γ, with

very little overlap between the two populations (Fig EV1K and L). Importantly, the activation defect was observed only in proinflammatory NK cells of *Calhm6*$^{-/-}$ mice, but not in the cytotoxic NK cell population (Fig EV1K and L). Collectively, these data suggest that CALHM6 is important at the very early stages of infection, where it affects the kinetics of NK cell activation for IFN-γ secretion. In the absence of CALHM6, innate responses are delayed, impairing the innate control of infection. Why early NK cell activation is defective in the absence of CALHM6 was tested next.

**Both *Calhm6*$^{-/-}$ macrophages and NK cells can respond normally to activating signals when stimulated in isolation *in vitro*, suggesting a defect in their cell–cell communication *in vivo***

Natural killer cells can be activated by contact-dependent signals delivered through the IS by myeloid cells, or by contact-independent

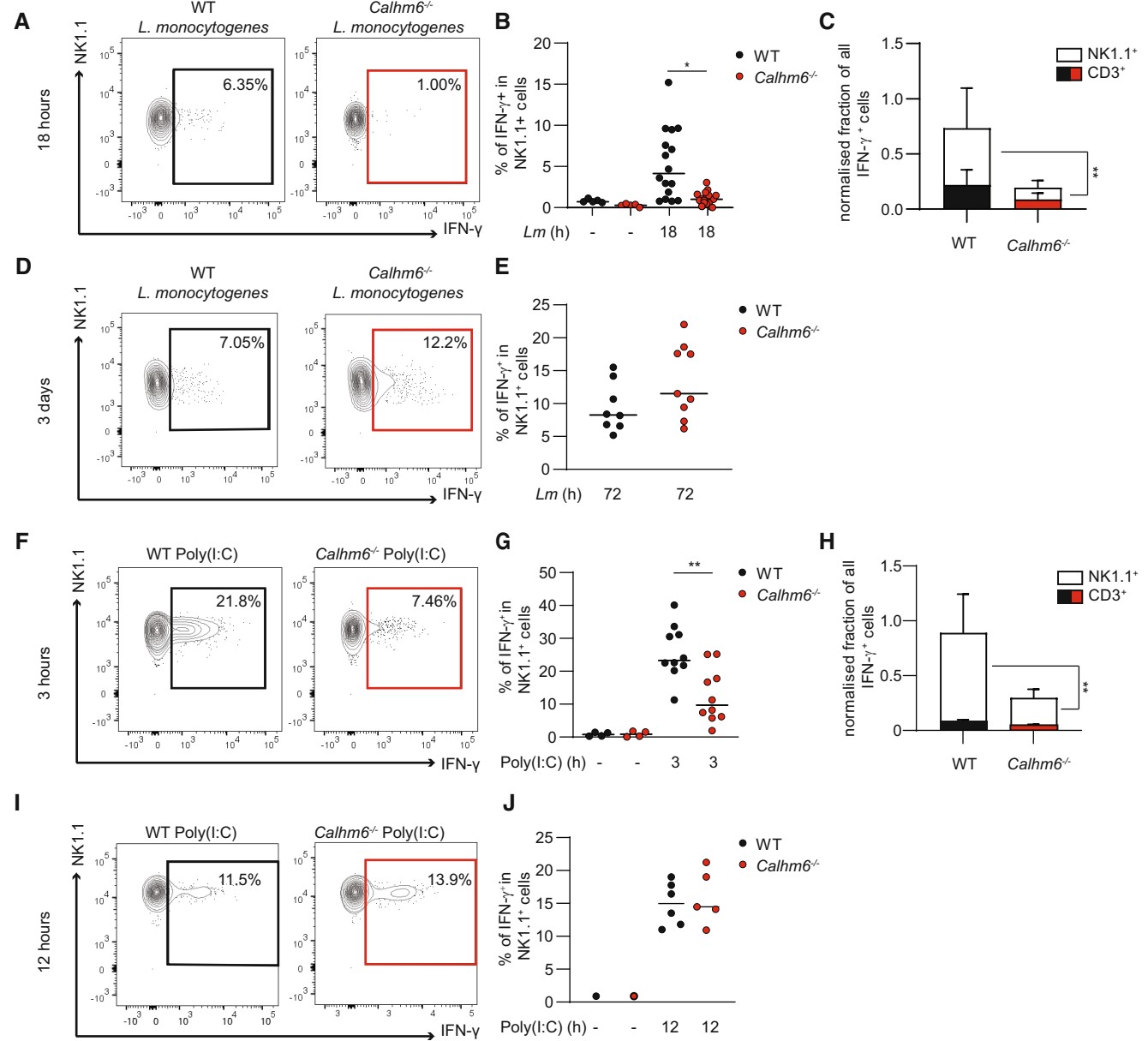

**Figure 3. Calhm6$^{-/-}$ mice have delayed NK cell activation in response to Listeria monocytogenes infection and to Poly(I:C) in vivo.**

A–E  WT and Calhm6$^{-/-}$ mice were injected IP with 3 × 10$^5$ CFU of L. monocytogenes or PBS. After 18 h (A–C) or 3 days (D, E) spleens were collected and splenocytes incubated for an additional 4 h with Brefeldin A before intracellular staining for IFN-γ and analysis by flow cytometry. (A, D) Representative contour plots, (B, E) pooled frequency of IFN-γ$^+$ cells in CD3$^-$NK1.1$^+$ cells gate from WT and Calhm6$^{-/-}$ mice, and (C) normalised to 1 IFN-γ$^+$ cells that are NK1.1$^+$ or CD3$^+$ based on WT, to pool experiments (For Lm 18 h: pooled results from four independent experiments, Poly(I:C) WT mice = 16, Calhm6$^{-/-}$ mice = 17, control WT mice = 5, Calhm6$^{-/-}$ mice = 5, One way ANOVA with multiple comparisons. For Lm 3 days: pooled results from two independent experiments, WT mice = 8, Calhm6$^{-/-}$ mice = 9, Mann–Whitney test).

F–J  WT and Calhm6$^{-/-}$ mice were injected i.p. with Poly(I:C) (200 μg/mouse) or PBS for 3 h (F–H) or 12 h (I, J), spleens were collected and processed as in (A–E). Representative contour plots (F, I), pooled frequency of IFN-γ$^+$ in CD3$^-$NK1.1$^+$ cells of WT versus Calhm6$^{-/-}$ mice (G, J) and ratio of IFN-γ$^+$ cells that are NK1.1$^+$ or CD3$^+$ normalised as in C (H) (For Poly(I:C) 3 h: pooled results from two independent experiments, Poly(I:C) WT mice = 10, Calhm6$^{-/-}$ mice = 10, control WT mice = 2, Calhm6$^{-/-}$ mice = 2, One way ANOVA with multiple comparisons. For Poly(I:C) 12 h: results from one experiment, Poly(I:C) WT mice = 6, Calhm6$^{-/-}$ mice = 5, control WT mice = 1, Calhm6$^{-/-}$ mice = 1 mice, One-way ANOVA with multiple comparisons).

Data information: *P < 0.05, **P < 0.01. Error bars represent SD.

soluble signals such as cytokines (Newman & Riley, 2007). To understand why Calhm6$^{-/-}$ NK cells have a delayed IFN-γ response at the early stages of infection, we first tested whether CALHM6

deficiency affected NK cell maturation. Murine NK cells can be divided into four subsets based on the expression of the surface markers CD27 and CD11b (Chiossone et al, 2009). Because CD27$^{high}$

NK cells are the most involved in cytokine secretion (Hayakawa & Smyth, 2006), we tested whether the lack of IFN-γ production in NK cells could be accounted for by the absence of this population. However, no differences in the abundance of the four NK-cell subsets were observed independent of genotype and stimulation (Fig 4A and B). We next tested the ability of *Calhm6*$^{-/-}$ NK cells to respond in isolation to *in vitro* exogenous soluble signals that are typically produced by macrophages and DCs in response to *L. monocytogenes* infection (D'Orazio, 2019). We treated FACS-sorted naïve CD3$^-$NK1.1$^+$ cells from WT and *Calhm6*$^{-/-}$ spleens with, IL-12, IL-18, and TNF. PMA/ionomycin served as a control (Fig 4C). There were no differences in the abilities of *Calhm6*$^{-/-}$ and WT NK cells to respond to soluble signals and produce IFN-γ. Even NK cells purified from mice previously injected with Poly(I:C) *in vivo* for 3 h, had nearly WT levels of IFN-γ production when stimulated with PMA/Ionomycin for an additional 4 h *in vitro* (Fig EV2A). These results indicate that CALHM6 is neither important for NK cell maturation nor required for NK cell activation by soluble factors. Hence, the observed *in vivo* defect could either result from an inability of accessory cells (macrophages or DCs) to produce soluble stimuli, or from their inability to engage in productive contact-dependent communication with NK cells at the IS.

Poly(I:C) *in vivo* induces strong Type I IFN responses in macrophages and DCs, leading to the transcription of IFN-stimulated genes (ISG) (Ivashkiv & Donlin, 2014). Thus, we measured the upregulation of ISGs (*Ifnar4*, *Ifnar14*, *Ifnb1*, *Il1r2*) in sorted splenic macrophages 3 and 6 h after Poly(I:C) injection, as a proxy of their ability to detect Poly(I:C) and produce systemic type I IFNs. We also directly measured the secretion of type I IFNs in the serum 1 and 3 h after Poly(I:C) injection, as well as the secretion of several other macrophage-derived inflammatory cytokines (TNF, IL-12, IL-1b, IL-6). We detected no differences in type-I IFN secretion or ISG induction between the genotypes (Fig EV2B–R) suggesting that *Calhm6*$^{-/-}$ macrophages can detect and respond to Poly(I:C) normally and that their early type I IFN response is not impaired.

To test whether macrophages could produce soluble signals required to activate NK cells during infection, we established an *in vitro* model to mimic *L. monocytogenes* infection in macrophages (see Materials and Methods) and measured IL-1β (Fig 4D) and IL-18 (Fig 4E) secretion in the supernatant of infected cells. These two cytokines are typically produced by *L. monocytogenes*-infected macrophages and are important for NK cell activation (Newman & Riley, 2007). Both WT and *Calhm6*$^{-/-}$ BMDM were capable of responding to *L. monocytogenes* infection *in vitro* and released similar levels of IL-1β and IL-18 (Fig 4D and E).

Finally, we tested if *Calhm6*$^{-/-}$ macrophages could kill *L. monocytogenes* when they are provided with exogenous IFN-γ. Both IFN-γ stimulation and bacterial infection induce nitric oxide synthase expression in macrophages to produce NO, an important cytotoxic effector against intracellular pathogens like *L. monocytogenes* (MacMicking *et al*, 1997; Pamer, 2004). A Griess test, which detects nitrite ions, was performed on supernatant of BMDM stimulated with either IFN-γ or bacterial ligands (Fig 4F). No differences in NO production by *Calhm6*$^{-/-}$ and WT BMDM were detected. Consistent with this result, both WT and *Calhm6*$^{-/-}$ BMDM similarly controlled *L. monocytogenes* replication *in vitro* when primed with IFN-γ prior to infection (Fig 4G).

We conclude that both NK cells and myeloid cells from *Calhm6*$^{-/-}$ mice are functional in isolation. First, macrophages can detect bacteria and Poly (I:C) and can secrete soluble cytokines essential for NK cells activation. Second, NK cells can respond to these cytokines, likely explaining why NK cells eventually become activated, albeit with delayed kinetics in *Calhm6*$^{-/-}$ mice. And third, macrophages can kill *L. monocytogenes* when they are provided with IFN-γ. Because the most efficient and fastest way to activate NK cells *in vivo* is via direct contact between activated myeloid cells and NK cells through the formation of an IS (Newman & Riley, 2007), we conclude that there is a deficit in the early contact-dependent communication between myeloid cells and NK cells in *Calhm6*$^{-/-}$ mice.

## CALHM6 in CD11c$^+$ dendritic cells is dispensable for NK cell activation and IFN-γ production

Because CALHM6 is also expressed, albeit at a much lower level, in activated DCs, and DCs were used in a previous *in vitro* study of CALHM6 (Ebihara *et al*, 2010) we tested the contribution of CALHM6 on DCs to NK cell activation *in vivo* in our inflammation model. We generated a CD11c-specific CALHM6 conditional KO mouse (Fig EV3A) and found that NK cells from *Itgax*$^{cre/-}$*Calhm6*$^{fl/fl}$ mice produced normal levels of IFN-γ and Granzyme B 3 h after *in vivo* Poly(I:C) stimulation (Fig EV3B–D). Based on these results and our finding that macrophages express CALHM6 at higher levels than in DCs, we conclude that CALHM6 expression in CD11c$^+$ cells is not required for NK cells activation *in vivo*.

To better understand the cell-type-specific role of CALHM6 in macrophages, DCs and NK cells, we tried to optimise a mixed *in vitro* co-culture system based on the one established by Kasamatsu *et al* (2014). We co-cultured either BMDCs, BMDM, or freshly isolated F4/80$^+$ cells from WT or *Calhm6*$^{-/-}$ spleens with either magnetically or FACS-sorted WT or *Calhm6*$^{-/-}$ NK cells, in the presence or absence of Poly(I:C). However, none of these co-culture conditions reliably recapitulated the delayed NK cell activation phenotype seen *in vivo* (Fig EV3E–G). A similar result was found using *Rag*$^{-/-}$*Calhm6*$^{-/-}$ mice, which do not develop adaptive immune cells, therefore avoiding any contribution of T cells to IFN-γ secretion (Fig EV3H). We hypothesised that any defects in contact-dependent activation of NK cells in *Calhm6*$^{-/-}$ co-cultures must happen quite early *in vitro*, and that accumulation of NK-activating soluble cytokines (e.g. IL-12, IL-18) over time in a small volume of the 96-well co-culture dish quickly compensates for the initial delay in activation, making the experiment subject to significant variability.

## CALHM6 is recruited to immunological synapses formed by activated macrophages

To determine if the absence of CALHM6 causes changes in macrophages that could explain their inability to communicate with NK cells, we performed an unbiased analysis of WT versus *Calhm6*$^{-/-}$ BMDM transcriptomes, using naïve cells, and cells stimulated for 24 h with Poly(I:C) *in vitro*. Very few differentially expressed genes (DEG) were observed between untreated WT and *Calhm6*$^{-/-}$ BMDM (Fig 5A), which was expected based on the very low/absent CALHM6 expression prior to activation (Fig 1). After Poly(I:C)

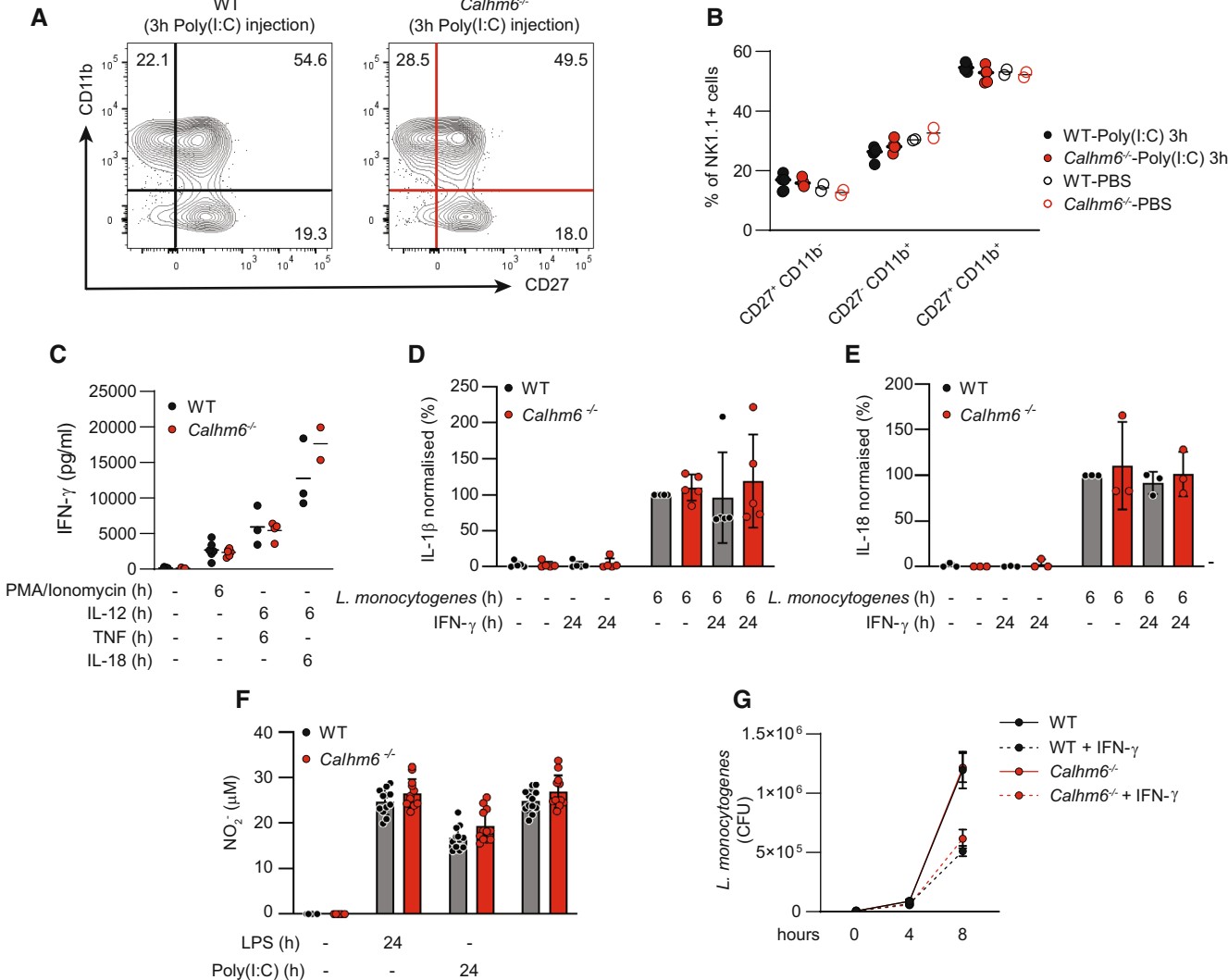

**Figure 4.** *Calhm6*⁻/⁻ **macrophages and NK cells are capable of normal responses to activating signals when stimulated in isolation, *in vitro*.**

A, B  WT and *Calhm6*−/− mice were injected i.p. with Poly(I:C) (200 μg/mouse) or PBS. On day 3 spleens were collected, stained with antibodies against NK cell-maturation markers and analysed by flow cytometry. Representative contour plots (A) and pooled flow cytometry results (B) are shown (results from one experiment, Poly(I:C) WT mice = 5, *Calhm6*−/− mice = 5, control WT mice = 2, *Calhm6*−/− mice = 2 mice, One-way ANOVA with multiple comparisons).

C  CD3⁻NK1.1⁺ cells were isolated from naïve mouse spleens and stimulated *in vitro* with PMA/ionomycin, IL-12 (50 ng/ml), TNF (50 ng/ml), or IL-18 (50 ng/ml). After 6 h stimulation, supernatant was collected and IFN-γ was measured by ELISA (pooled results from two independent experiments, Untreated and PMA stimulation WT mice = 6, *Calhm6*−/− mice = 6, TNF + IL-12 stimulation: WT mice = 3, *Calhm6*−/− mice = 3, otherwise WT mice = 3, *Calhm6*−/− mice = 2, One-way ANOVA with multiple comparisons).

D, E  BMDM with or without 24 h IFN-γ priming were grown in antibiotic-free medium, washed and cultured with *Listeria monocytogenes* (MOI 1) for 30 min, at which time gentamicin (5 ng/ml) was added to the medium to kill extracellular bacteria. Supernatant from BMDM infected with *L monocytogenes* was collected after 6 h. IL-1β (D, pooled results from five independent experiments, WT mice = 5, *Calhm6*−/− mice = 5, Kruskal–Wallis test) and IL-18 concentration (E, pooled results from three independent experiments, WT mice = 3, *Calhm6*−/− mice = 3, Kruskal–Wallis test) were quantified by ELISA, and normalised to WT set to 100%, to allow pooling of independent experiments.

F  NO production in WT and *Calhm6*−/− BMDM treated overnight with LPS or Poly(I:C) quantified with a Griess test on cell supernatant (results from one experiment, WT mice = 3, *Calhm6*−/− mice = 3, Kruskal–Wallis test with multiple comparisons).

G  BMDM with or without IFN-γ priming were grown overnight in antibiotic-free medium. Cells were infected with *L monocytogenes* (MOI 1) for 30 min. Gentamicin (5 ng/ml) was then added to the medium to kill extracellular bacteria. At 0, 4 and 8 h cells were harvested, lysed with 0.1% Triton X-100 and the lysate was plated on antibiotic-free BHI plates for 24 h, and CFU were quantified (pooled results from three independent experiments, WT mice = 3, *Calhm6*−/− mice = 3, Two-way ANOVA test).

Data information: Error bars represent SD.
Source data are available online for this figure.

stimulation, *Calhm6*$^{-/-}$ BMDM failed to upregulate the expression of several genes (Fig 5B), with enrichment for genes associated with synaptic pathways traditionally linked with neuronal synapses (Fig 5C). This was true not only for Poly(I:C) stimulated BMDM but also for BMDM treated with IFN-γ, LPS, LPS + IFN-γ, Zymosan, and Zymosan + IFN-γ, that is in response to all the pro-inflammatory signals known to upregulate *Calhm6* (Dataset EV1). The genes that failed to be upregulated in *Calhm6*$^{-/-}$ BMDM are involved in such functions as vesicle trafficking and fusion (*Cplx1/2*, *Snph*, *Rph3a* and *Clstn2*) (Kato *et al*, 1996; Lao *et al*, 2000; Kang *et al*, 2008a; Trimbuch & Rosenmund, 2016; Ranneva *et al*, 2020), and formation of synaptic contacts (*Map1a* and *b*, *Cntn1* and *Obsl1*) (Szebenyi *et al*, 2005; Geisler *et al*, 2007; Yan *et al*, 2013; Veny *et al*, 2020). These data implicate CALHM6 in synapse formation in BMDM, possibly by regulating vesicle trafficking and/or fusion. The detection of DEGs in pure BMDM cultures also suggests that even before contact with NK cells, there is low-level autocrine function of CALHM6 in these cells, possibly because of the formation of macrophage-to-macrophage cell–cell contact. However, the overall number of DEG in all conditions tested was modest in these pure BMDM cultures, suggesting that the main role of CALHM6 is likely in the paracrine macrophage-to-NK communication rather than in autocrine macrophage-to-macrophage signalling.

To test whether CALHM6 has a role in cell contact-mediated activation of NK cells, we established an *in vitro* model to visualise CALHM6 localisation in activated macrophages and NK cells. Because we found that none of the commercially available CALHM6 antibodies were suitable for immunolocalisation, we generated C-terminally tagged CALHM6 (CALHM6-FLAG and CALHM6-eGFP) and expressed them in BMDM using retroviral transduction. Steady-state localisation revealed that most of the ectopically expressed CALHM6 was intracellular (Fig EV4A and B), irrespective of whether the small FLAG or a large eGFP-tagged CALHM6 was used. To visualise the IS, we sorted fresh NK cells from spleens of WT mice and added them on top of a CALHM6-eGFP transduced BMDM monolayer, previously primed with IFN-γ and LPS for 6 h. NK cells were easily distinguished by cell size and morphology (white arrow in Fig 5C). IS sites were imaged by serial Z-sections; a collapsed Z-stack is shown in Fig 5D. We quantified CALHM6 proximity to the IS of activated BMDMs 10, 20 and 40 min after NK-cell addition using only the Z-slice that contained the NK cell-BMDM contact site (Fig 5E). At *t* = 0 min, CALHM6 was not localised near the IS in any

of the contacts imaged. The percentage of imaged IS that showed recruitment of CALHM6 towards NK cell IS increased significantly with time, until 40 min when about 50% of all actin-ring positive contacts had also recruited CALHM6-eGFP from the BMDM cytoplasm towards the NK cell (Fig 5E).

To test whether the recruitment of CALHM6 towards the BMDM synapse was unique to NK cells or if CALHM6 is also recruited to other BMDM synapses, we used our *in vitro* model to visualise the BMDM phagocytic synapse, formed with either PKH26 stained sheep red blood cells (sRBC) opsonised with anti-sRBC-IgG2a (Fig 5F and G), non-opsonised sRBC (Fig EV4C and D), or non-opsonised 3 μm polystyrene beads (Fig EV4E and F), where only a single Z-slice containing sRBC contact site with BMDM is shown. For phagocytosis to happen, a structure very similar to the IS needs to be formed, with actin remodelling around the target (Niedergang *et al*, 2016). As in the BMDM-NK cell synapse, CALHM6 was also recruited towards the phagocytic synapse in a time-dependent manner. These results indicate that CALHM6 is recruited to the plasma membrane of BMDM as a part of the synapse formed with either phagocytic cargo or other immune cells.

Because CALHM6 recruitment to the synapse might affect the phagocytic process, we tested whether BMDM require CALHM6 for efficient phagocytosis. For this, phagocytic rates for opsonised and non-opsonised sRBCs (Fig EV4G), Zymosan particles (Fig EV4H) or endocytosis rates of Dextran 10,000 (Fig EV4I) were measured. No genotype-dependent differences in the phagocytic or endocytic rates were observed, suggesting that CALHM6 is dispensable for these processes in BMDM.

**CALHM6 forms an ion channel in the plasma membrane of *Xenopus* oocytes, but is sequestered in intracellular compartments of mammalian cells prior to activation**

Cryo-EM analysis of human placental CALHM6 showed that it forms a large channel-like structure composed of 10 or 11 monomers (Drozdzyk *et al*, 2020). Because the structure of endogenous CALHM6 in immune cells has not been determined, we first tested whether CALHM6 in BMDM also forms a large supramolecular complex. For this, we lysed IFN-γ or LPS stimulated BMDM in non-denaturing conditions and resolved the proteins on a Native Blue-PAGE gel (Fig 6A). Under non-denaturing conditions, CALHM6 resolves around 720 kDa, suggesting that endogenous CALHM6 is

---

**Figure 5. CALHM6 is recruited to synapses formed by activated BMDMs.**

A–C    BMDM were left untreated or stimulated for 24 h with Poly(I:C) (10 μg/ml) before RNA extraction and bulk RNASeq analysis. (A, B) Volcano plot of differentially expressed genes between untreated (A) or Poly(I:C) stimulated (B) WT and *Calhm6*$^{-/-}$ BMDM. (C) Bubble plot of Gene Ontology analysis of DEGs from poly(I:C) stimulated WT and *Calhm6*$^{-/-}$ BMDM (results from one experiment, WT mice = 3, *Calhm6*$^{-/-}$ mice = 3).

D–G    BMDM were transduced with a retroviral vector expressing CALHM6-eGFP. BMDM ($3 \times 10^5$) were plated on coverslips and pre-treated with IFN-γ and LPS for 6 h. (D) Freshly isolated primary CD3$^-$NK1.1$^+$ cells isolated from mouse spleens were added for 0, 10, 20, or 40 min on top of plated BMDMs that were generated from bone marrow cells transduced with a retroviral vector expressing CALHM6-eGFP. The interacting cells were then fixed and stained with phalloidin to identify actin and DAPI to identify nuclei. Synaptic interaction between BMDM and NK cell is indicated by white arrowheads; entire collapsed imaging Z-stack is shown. (E) Images were blinded and CALHM6-eGFP polarisation towards CD3$^-$NK1.1$^+$ cells manually scored only in Z-slices where BMDM-NK interaction was identified (*P*-value calculated by two-way ANOVA of polarisation and time points). Numbers on top of each bar indicate number of synapses scored for that time point. (F) IgG2a-opsonised sRBC labelled with PHK26 were added on top of plated CALHM6-eGFP-expressing BMDM. One single Z-slice containing BMDM-sRBC interaction plane is shown. Synaptic interaction between BMDM and sRBC cell is indicated by white arrowheads (G) Quantified results of experiment in F. Numbers on top of each bar indicate number of synapses scored for that time point. Longer time points were not possible to score for opsonised sRBC because of the speed of the phagocytosis as sRBC were quickly internalised. Images taken with 60× objective (*P*-value calculated by two-way ANOVA of polarisation and time points).

Source data are available online for this figure.

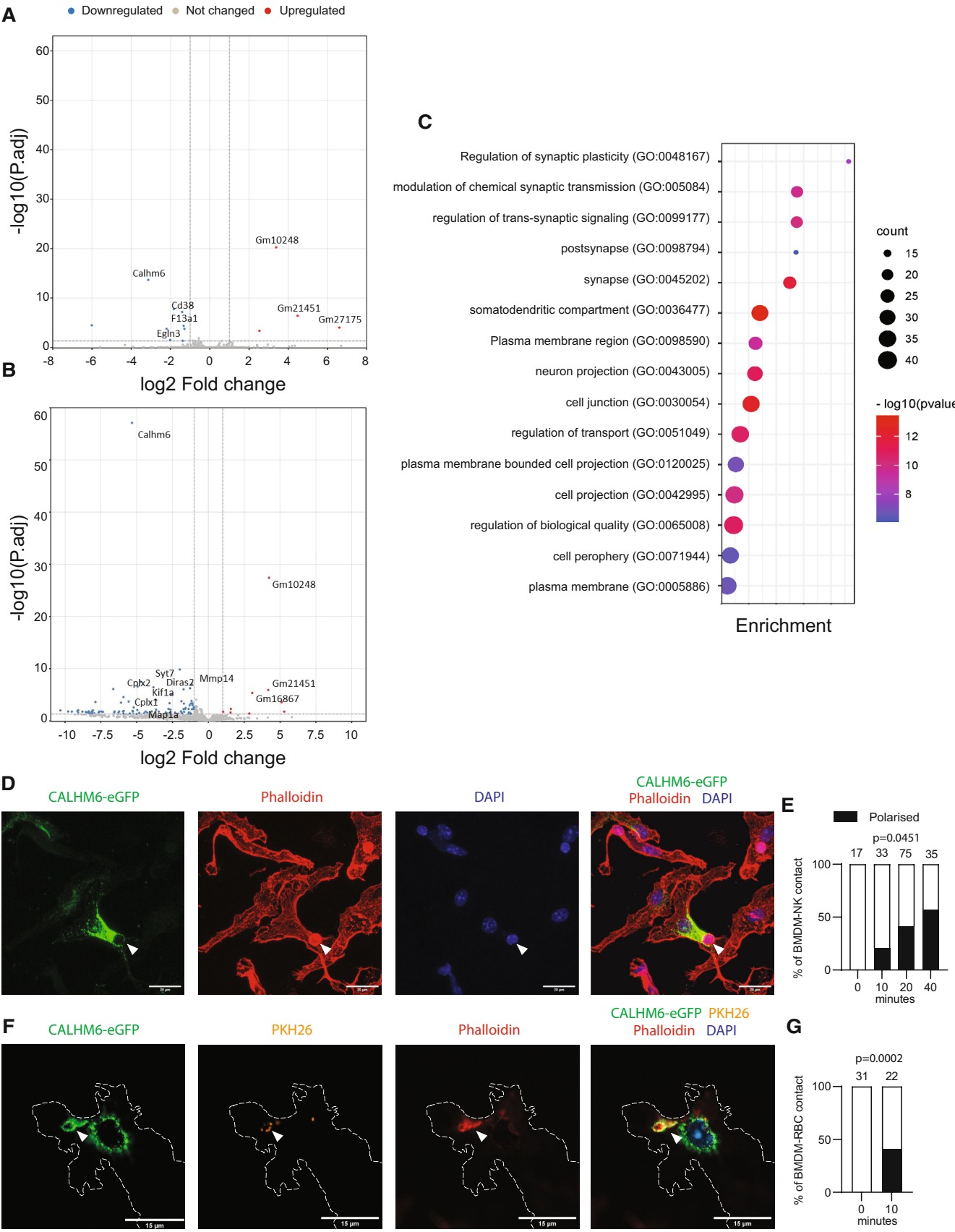

Figure 5.

part of a larger complex in BMDM, significantly larger than the CALHM1 complex, which contains eight subunits. As disulphide bonds in the inter-helix loop are important for the integrity of other CALHM channels (Siebert *et al*, 2013; Yang *et al*, 2020; Kwon *et al*, 2021), we tested if this was true for CALHM6 as well, by resolving the non-reduced protein by SDS–PAGE with increasing concentrations of DTT (Fig 6B). Like in CALHM1, the presence of a reducing agent broke down the CALHM6 complex to its monomeric constituents. Hence, at least biochemically, CALHM6 behaves like other CALHMs.

To determine whether CALHM6 can form an ion channel, we used biophysical approaches previously used to characterise CALHM1 (Ma *et al*, 2012). CALHM1 forms a large-pore, voltage-gated ATP-permeable non-selective channel that is activated by the reduction of extracellular $Ca^{2+}$ concentration and by membrane depolarisation (Sana-Ur-Rehman *et al*, 2017; Ma *et al*, 2018). To test whether mouse CALHM6 (mCALHM6) can also form a plasma membrane ion channel, we expressed it in *Xenopus* oocytes and in the mouse N2a cell line, as we previously did for CALHM1 (Ma *et al*, 2012). For imaging, we used tagged mCALHM6-GFP, while for all other functional assays, we used untagged protein. In oocytes, tagged WT-mCALHM6-GFP locates to the plasma membrane (Fig 6C). Expression of the untagged WT-mCALHM6 was highly toxic, causing cell lysis and death in a Calhm6-mRNA dose-dependent manner (Fig 6D). CALHM1 is also (albeit much less) toxic to oocytes as a result of its finite open probability and large pore. Currents through the CALHM1 channel can be prevented by mutations of a conserved acidic residue (D121) near the extracellular end of the third transmembrane helix (Ma *et al*, 2012). To determine whether toxicity associated with CALHM6 expression was due to its ability to function as an ion channel, we expressed CALHM6 with the same conserved acidic residue mutated. Oocytes expressing E119R-mCALHM6 remained as viable as control-injected cells. Notably, like WT-CALHM6, E119R-mCALHM6 was similarly expressed at the plasma membrane (Fig 6C and D). These results suggest that CALHM6 functions as an ion channel when it is expressed in *Xenopus* oocytes where its non-zero open probability results in cell death. To directly determine whether CALHM6 expression was associated with a novel conductance in oocytes, we performed two-electrode voltage clamp of oocytes expressing very low levels of mCALHM6 that prevent cell toxicity. In response to step changes in the plasma membrane voltage from −80 mV to +70 mV in 10-mV intervals from a holding potential of −15 mV, large outwardly rectifying currents were observed in a bath solution containing 2 mM divalent cations. These currents were mediated specifically by CALHM6 since they were $Gd^{3+}$-sensitive, and similar currents were not observed in control-injected oocytes or, importantly, in cells expressing E119R-mCALHM6 (Fig 6F). Of interest, mCALHM6 channel currents were not fully inhibited in bath solutions containing mM divalent cations (2–10 mM $Ca^{2+}$ or $Ba^{2+}$), which is distinct from extracellular $Ca^{2+}$-dependent closing of CALHM1 and CALHM1/3 channels. This distinction likely accounts for the pronounced toxicity of CALHM6 compared with CALHM1. That is, constitutive leak currents through mCALHM6 in normal divalent cation-containing bath solutions resulted in cell death, especially pronounced in oocytes expressing high levels of mCALHM6 and mitigated by abolishing CALHM6 currents by a mutation in E119R-mCALHM6 (Fig 6G). Collectively, these results indicate that CALHM6 functions as a voltage-sensitive plasma membrane ion channel in *Xenopus* oocytes. Interestingly, when expressed in N2a cells, WT-mCALHM6-GFP was largely sequestered in intracellular compartments (Fig 6E), similar to our observations in resting macrophages (Fig EV5A and B). Thus, mammalian cells appear to restrict CALHM6 localisation to intracellular compartments, from where, at least in macrophages, it gets recruited to the surface (Fig 5D and F) upon macrophage activation and synapse formation.

## CALHM6 channel facilitates ATP release from activated macrophages but, unlike other CALHMs, reduction of extracellular $Ca^{2+}$ is not sufficient to open CALHM6

Acute removal of extracellular $Ca^{2+}$ activates CALHM1 channels. This can be observed by monitoring changes of cytoplasmic $Ca^{2+}$ concentration upon $Ca^{2+}$ add-back to the bathing solution, when $Ca^{2+}$ can flow into the cell through open CALHM1 channels (Fig 7A) (Siebert *et al*, 2013; Taruno *et al*, 2013; Ma *et al*, 2018), In contrast, $Ca^{2+}$ removal and subsequent "add-back" was without effect on

**Figure 6. CALHM6 forms a plasma membrane ion channel.**

A   BMDM were stimulated for 24 h with IFN-γ (10 ng/ml) or LPS (100 ng/ml) and harvested in 1% digitonin or 1% DDT buffer, with cOmplete Protease Inhibitor Cocktail. The lysate was resolved by Native BluePAGE gel under non-denaturing conditions, and probed with anti-CALHM6 antibody (one representative experiment of two is shown, *n* = 3 mice).

B   WT or Calhm6⁻/⁻ BMDM stimulated with IFN-γ (10 ng/ml) for 24 h were lysed with 1% Triton X100-containing buffer with cOmplete Protease Inhibitor Cocktail and treated with increasing concentrations of DTT. The samples, without boiling, were resolved by SDS–PAGE gel (one representative experiment of two is shown, *n* = 4 mice). Lack of DTT influences solubilisation of GADPH monomer, and its detection for immunoblotting.

C   WT-mCALHM6-GFP and E119R-mCALHM6-GFP localise to the plasma membrane of *Xenopus* oocytes. Upper: transmitted light image; Lower: confocal fluorescence image at oocyte equator (Images representative of three independent experiments, oocytes = 6).

D   Expression of WT-mCALHM6 results in oocyte death 24 h after injection of either 5 ng (left) or 50 ng (middle) mCALHM6 cRNA, but oocytes injected with 50 ng E119R-mCALHM6 cRNA remain healthy (right) (Images representative of three independent experiments, oocytes = 5–10).

E   Localisation of mCALHM6-eGFP in N2a cells 48 h post-transfection. mCALHM6 localises predominately in intracellular compartments (Images representative of three independent experiments, *n* = 7 cover glasses analysed).

F   Representative families of currents recorded by two-electrode voltage-clamp in *Xenopus* oocytes, evoked by 3-s voltage pulses from −80 mV to +70 mV in 10-mV intervals from a holding potential of −15 mV in 2 mM $Ba^{2+}$ bath solution. Left: currents in oocyte injected with WT-mCALHM6 cRNA; red current trace recorded at −80 mV from −15 mV holding potential after addition of 2 mM $Gd^{3+}$; Middle: currents in oocyte expressing E119R-mCALHM6; Right: currents in oocyte injected with connexin 26 antisense oligonucleotide (ASO) only (control). Dashed line: zero-current level.

G   Current–voltage (I-V) relations for WT-mCALHM6, E119R-mCALHM6 and control (ASO) in 2-mM $Ba^{2+}$ bath solution (pooled individual whole-cell patches, WT-mCALHM6 *n* = 13, E119R-mCALHM6 *n* = 4, control = 4, SEM error bars).

Source data are available online for this figure.

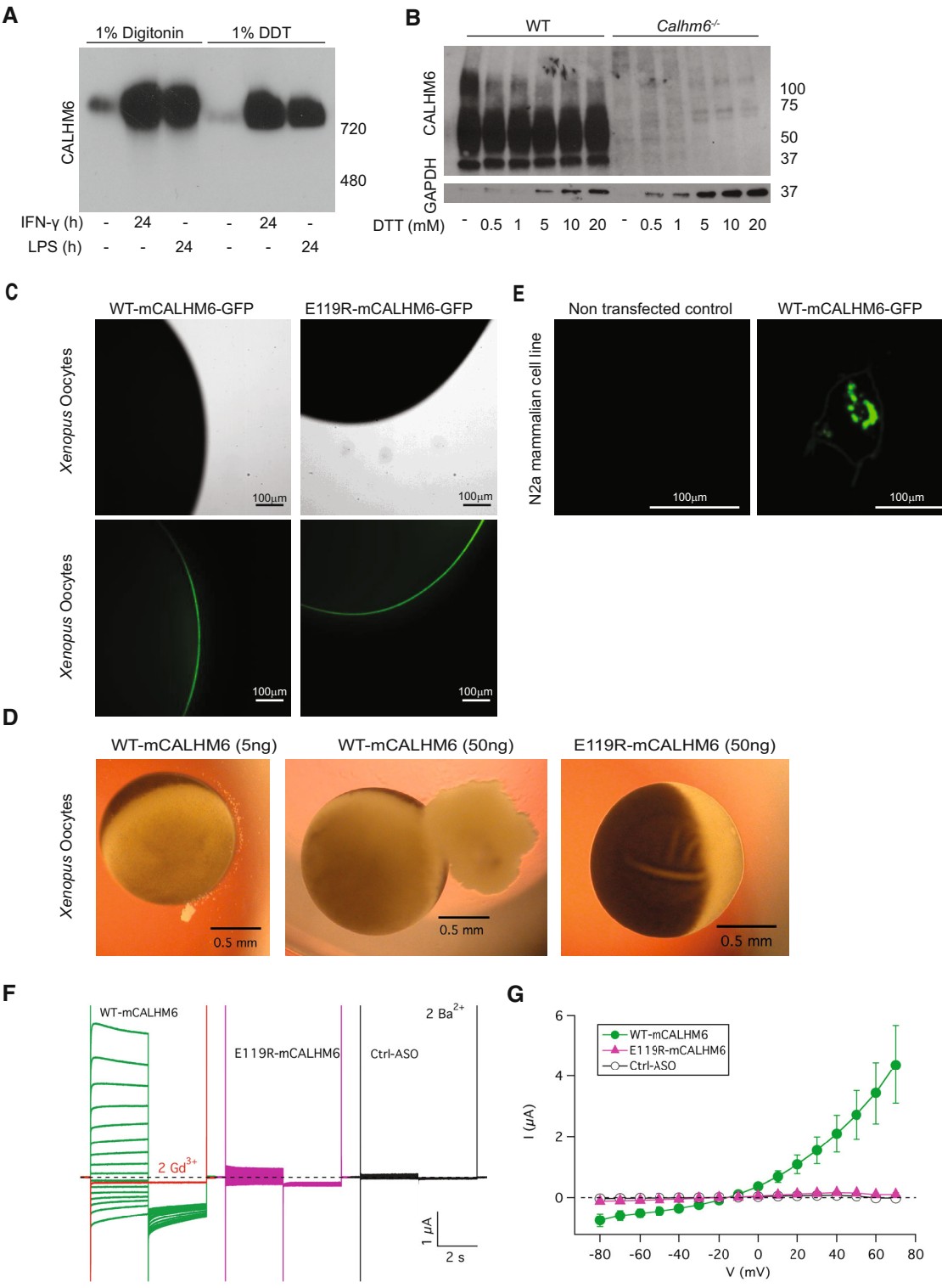

Figure 6.

cytoplasmic $Ca^{2+}$ concentration in CALHM6-expressing N2A cells (Fig 7A). A similar protocol using WT and $Calhm6^{-/-}$ BMDMs also failed to reveal an extracellular-$Ca^{2+}$ dependent CALHM6 permeability (Fig 7B). In addition, acute removal of extracellular $Ca^{2+}$ for 20 min did not induce ATP release from BMDMs (Fig 7C). As expected from

the known metabolic changes in macrophages upon activation (Wang et al, 2018), ATP release was somewhat lower in activated BMDMs than in naïve cells but, importantly, it was independent of CALHM6 in either condition (Fig 7C). In contrast, expression of CALHM1, but not CALHM6, facilitated ATP release from HEK293T cells after acute

removal of extracellular $Ca^{2+}$ for 20 min (Fig 7D). These results are consistent with an apparent absence of CALHM6 in the plasma membrane of mammalian cells. To determine if stimulation-induced CALHM6 translocation to the membrane resulted in a CALHM6-dependent ATP permeability, WT and CALHM6 KO macrophages were stimulated for 3 h with poly(I:C) and LPS to induce maximal CALHM6 expression. The cells were maintained in divalent cation-containing culture media in the presence of ATPase inhibitors to prevent hydrolysis of any released ATP. Notably, we detected steady, CALHM6-dependent, ATP release only from activated WT but not $Calhm6^{-/-}$ BMDMs (Fig 7E). CALHM2 was reported to act as a chaperone that regulates membrane trafficking and signalling of the ATP receptor and channel P2X7 (P2X7R) (Cheng *et al*, 2021). However, CALHM6 did not affect either P2X7R expression on BMDMs (Fig EV5A and B) or ATP-P2X7R dependent signalling as measured by ATP-induced inflammasome activation (Fig EV5C–E). Accordingly, we conclude that the observed CALHM6-depednet ATP release was mediated directly by CALHM6 localised in the IS. Our results are most consistent with a model in which CALHM6 mediates ATP release from activated macrophages that requires several steps: first, CALHM6 expression is upregulated upon macrophage activation, second, CALHM6 is recruitment to the plasma membrane at the immunological synapse, and third, CALHM6 channel opening is enhanced, possibly by local signals in the IS. Such stepwise control may be necessary in mammalian cells to prevent toxicity observed in CALHM6-expressing oocytes.

## Discussion

In this study, we generated $Calhm6^{-/-}$ mice and showed that they fail to control *L. monocytogenes* burden at the peak of infection due to delayed kinetics of the innate immune response. In $Calhm6^{-/-}$ mice, IFN-$\gamma$ production in NK cells was nearly absent in the early phases of the immune response to *L. monocytogenes* and Poly(I:C), but $Calhm6^{-/-}$ mice could produce normal, albeit very delayed IFN-$\gamma$ response. Consequently, they were also delayed in the induction of an anti-inflammatory feedback loop and IL-10 secretion. The

delayed kinetics of innate responses had a domino effect on the ability of $Calhm6^{-/-}$ mice to control bacterial burden in the early stages of infection. In $Calhm6^{-/-}$ mice the pathogen does not initially encounter IFN-$\gamma$-primed macrophages and so it is not well controlled, which together with delayed inflammation makes the mice more sensitive to the infection (Synopsis figure). The functional defect is not due to an intrinsic failure of macrophages to kill *L. monocytogenes*, as IFN-$\gamma$ provided *in vitro* rescued the defect. Neither was the defect due to an intrinsic failure of NK cells to produce IFN-$\gamma$, as providing soluble NK-activating cytokines *in vitro* rescued it. Rather the inability to control *L. monocytogenes* was due to the failure of CALHM6-deficient macrophages to activate NK cells and elicit timely IFN-$\gamma$ production *in vivo*.

Following *L. monocytogenes* infection, IFN-$\gamma$ production peaks around 20–24 h, driven almost entirely by NK cells during this period (Kang *et al*, 2008b; Kubota & Kadoya, 2011; Clark *et al*, 2016) Lack of IFN-$\gamma$ signalling to myeloid cells in this window manifests itself 72 h later with increased bacterial burden (Lee *et al*, 2013) as seen in our model. 3 days post-infection, NK cells are a major source of IL-10 (Perona-Wright *et al*, 2009; Valderrama *et al*, 2017; Clark *et al*, 2018); these NK cells have switched from releasing IFN-$\gamma$ to IL-10, and rarely produce both at the same time (Clark *et al*, 2016). The timing of this switch is important for balancing the response to infection and immunopathology: a series of studies from the Lenz group (Humann *et al*, 2007; Clark *et al*, 2016, 2018) and others (McCullers, 2014; Liu *et al*, 2019) have shown that NK cell-derived IL-10 makes mice more susceptible to infection if produced too early. This requirement for an exact regulation of the kinetics of the immune response is not a prerogative of *L. monocytogenes* infections. Timing of CALHM6-dependent IFN-$\gamma$ expression could be of particular relevance in controlling the magnitude of the immune response to co-infection or in chronic inflammation. Primary exposure to viral infection, or chronic inflammatory conditions, often sensitises mice for excessive response to secondary immune challenges and can have detrimental outcomes if the overlap is too close.

We have demonstrated that CALHM6 expression is highest in macrophages, that it is transient during infection and under the tight

---

**Figure 7. CALHM6 channel facilitates ATP release from activated macrophages.**

A  Fura-2 fluorescence ratio (F340/380) recorded in N2a cells expressing human CALHM1 (hCALHM1), mouse CALHM6 (mCALHM6) or control (vector) in response to removal and subsequent add-back of extracellular $Ca^{2+}$. Traces representative of four independent experiments for each, normalised to baseline ratio. Right: Resting F340/380 ratio in N2a cells expressing hCALHM1, mCALHM6 or vector in 2 mM $Ca^{2+}$ (the error bars represent SEM).

B  F340/380 ratio in response to $Ca^{2+}$ add-back protocol in mouse BMDMs with and without stimulation by Poly(I:C) and LPS. Traces representative of four independent experiments for WT macrophages and three for $Calhm6^{-/-}$ macrophages. Right: Resting F340/380 ratio in WT and CALHM6-KO macrophages in 2 mM $Ca^{2+}$ solution (the error bars represent SEM).

C  Naïve or overnight IFN-$\gamma$-primed BMDM from WT and $Calhm6^{-/-}$ mice were exposed to isotonic medium containing 5 mM of EDTA (to chelate extracellular $Ca^{2+}$) and 1 mM ATPase inhibitor ARL67156 (to prevent degradation of any released ATP) for 20 min. ATP concentration in the medium was measured with ENLITEN Luciferase/rLuciferin assay (pooled results from two independent experiments, WT mice = 4, $Calhm6^{-/-}$ mice = 4, one-way ANOVA with multiple comparisons).

D  Plat-E cells were transfected with IRES-GFP-expressing plasmids containing only a HA-tag, CALHM6-HA, or CALHM1-HA. $Ca^{2+}$ concentrations were acutely changed for 20 min in isotonic medium containing either EDTA or excess $K^+$. ATP release was measured as described above (pooled results from three independent experiments, $Calhm6^{-/-}$ mice = 3, error bars represent SD).

E  Steady-state ATP release from activated macrophages: BMDMs were plated at 150,000 cells/well in a 48-well plate, and incubated for 3 h with Poly(I:C) (50 µg/ml), LPS (100 ng/ml) and ATPase inhibitor ARL67156 (1 mM) in culture medium. Control cells were incubated with 1 mM ARL67156 only. ATP release (nM) measured using Sigma ATP Bioluminescent Assay: WT without stimulation: 9.12 ± 0.76; WT with stimulation: 22.16 ± 1.83; $Calhm6^{-/-}$ cells without stimulation: 8.76 ± 0.55; KO cells with stimulation: 12.19 ± 0.75 (pooled results from three independent experiments, WT mice = 3, $Calhm6^{-/-}$ mice = 3, Two-tailed unpaired Student's *t*-test, error bars represent SEM).

Data information: **$P < 0.01$, ***$P < 0.001$.
Source data are available online for this figure.

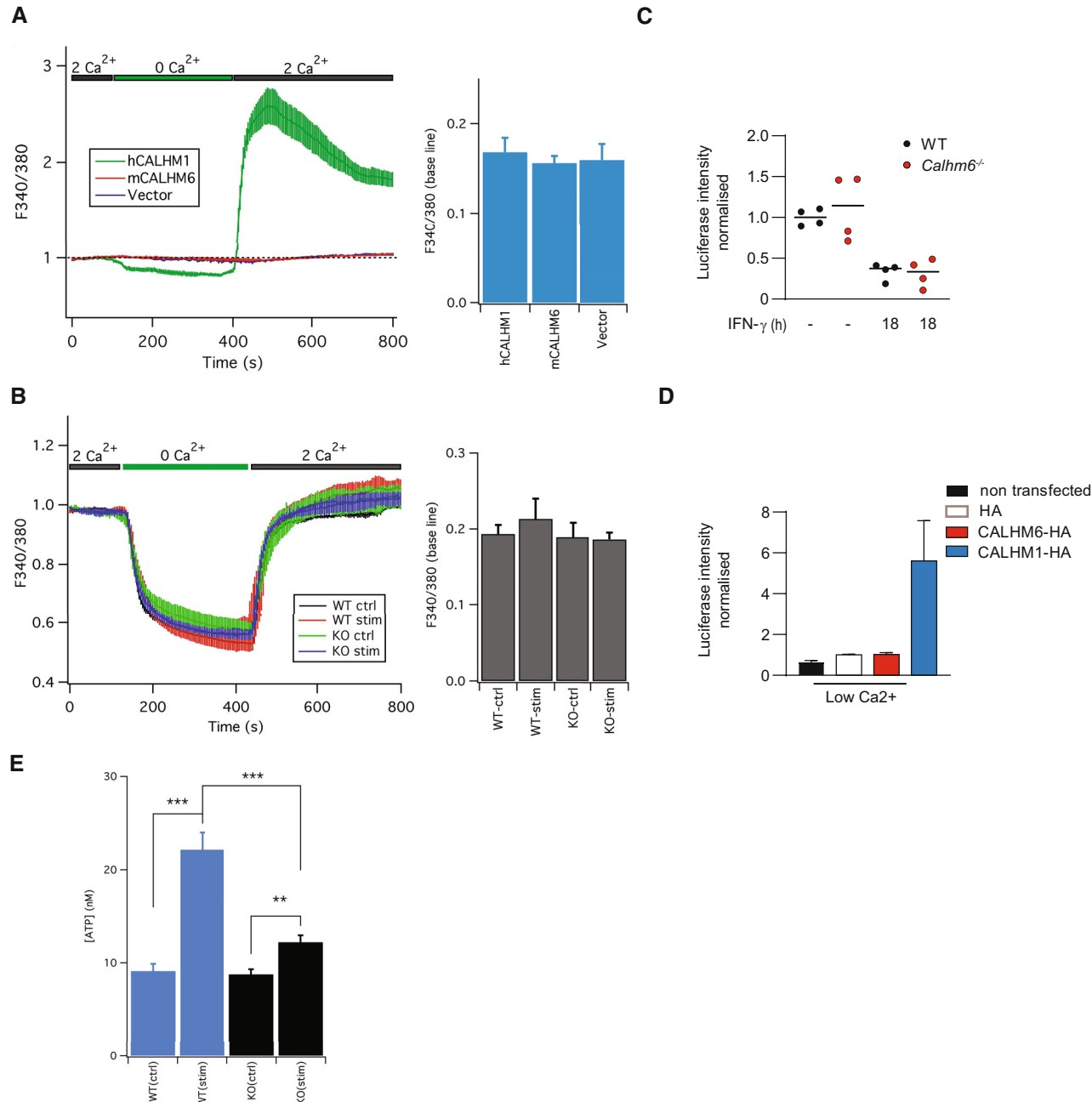

**Figure 7.**

control of pro- and anti-inflammatory signals. CALHM6 is induced in macrophages in response to IFNs and pathogen-derived signals, and its expression was stronger and longer if IFN-γ was also present. Like many genes involved in the initiation of inflammation, CALHM6 expression was also strongly downregulated by cytokines associated with the resolution of inflammation and tissue repair, such as IL-10, IL-4 and TGF-β. Thus, CALHM6 behaves as a typical early pro-inflammatory protein in macrophages involved in initiating the activation of other innate effector cells. Our results strongly suggest that CALHM6 plays a critical role in macrophages in their activation of NK cells via synaptic communication. The macrophage/NK cell

synapse is not as well studied as the IS between DC-T cells or the cytotoxic synapse (Newman & Riley, 2007; Dustin & Long, 2010). Increasingly promising results in NK cell-based cancer immunotherapy (Chiossone et al, 2018), highlight the need for a better understanding of cell-mediated NK cell activation in general, and the role of macrophages in this process in particular.

Mechanistically, we show that CALHM6 is upregulated by pathogen-derived signals on macrophages, and it translocates from intracellular compartments to the IS formed by activated macrophages and NK cells (Synopsis figure). Here, CALHM6 mediates the release of ATP (and possibly other signalling metabolites) from

activated macrophages. These results suggest that a functional role for CALHM6 reminiscent of that of CALHM1/3 at taste-bud cell-gustatory neuron synapse (Romanov *et al*, 2018) where the CALHM1/3 channel releases ATP as a neurotransmitter into the synaptic cleft. In both neuronal and immune synapses, ATP may be generated by mitochondria that are recruited to the synapse to provide local energy support for enhanced signalling demands. When released, synaptic ATP is detected by purinergic receptors, such as P2X7. Whether ATP released by macrophages in the IS acts on NK cells or back on macrophages in an autocrine fashion, to facilitate synaptic communication, remains to be determined. Our RNASeq results revealed that activated *Calhm6*$^{-/-}$ BMDMs failed to upregulate many of the genes involved in synaptic cell–cell communication, suggesting some autocrine CALHM6 contribution to macrophage activation. Interestingly, the genes we identified are traditionally investigated in neuronal synapses. Nonetheless, basic cell biological functions are conserved between neuronal and immune synapses (Dustin & Colman, 2002; Newman & Riley, 2007; Dustin & Long, 2010). For example, IL-12 is accumulated in vesicles that are released into the immune synapse cleft, akin to traditional neurotransmitter release at neuronal synapses, and myeloid IL-18 vesicles polarise towards NK cells after the formation of the DC-NK cell immunological synapse (Borg *et al*, 2004; Semino *et al*, 2005). Like neuronal synapses, several proteins traffic and cluster at the cell–cell interface in the IS (Piccioli *et al*, 2002; Semino *et al*, 2005). Furthermore, the actin cytoskeleton is remodelled at the IS (Borg *et al*, 2004), promoting fusogenic SNARE proteins localisation at the IS, similar to their localisation in neuronal synapses (Piccioli *et al*, 2002). We believe that CALHM6 is likely not important for establishing the initial IS cell–cell contact, nor does it act as a receptor for NK cell ligands. Instead, our results suggest that CALHM6 is important for the localised release of activating neurotransmitter-like factors, for example ATP at the IS (Romanov *et al*, 2018).

Cryo-electron microscopy of recombinant CALHM6 revealed that it assembles into a large-pore channel (Drozdzyk *et al*, 2020). However, the function of CALHM6 as an ion channel has not been previously demonstrated. Our biochemical results indicate that BMDM. CALHM6 monomers interact in a large oligomeric complex requiring disulphide bonds for its stability, consistent with the structural observations of recombinant CALHM6 (Demura *et al*, 2020; Drozdzyk *et al*, 2020). We determined that when expressed in *Xenopus* oocytes, CALHM6 localised to the plasma membrane and formed a voltage-gated ion channel with pharmacological properties, including inhibition by $Gd^{3+}$ and ruthenium red, similar to CALHM1 and CALHM1/3 channels expressed in this system. Mutation of a conserved acidic residue that inhibits CALHM1 channels blocked the CALHM6-induced conductance without affecting its plasma membrane localisation in oocytes, indicating that the observed currents were mediated by CALHM6. Inhibition of CALHM6 conductance by extracellular divalent cations, that is $Ca^{2+}$, $Ba^{2+}$, is less effective than in CALHM1 and CALHM1/3 channels, resulting in a constitutive CALHM-mediated plasma membrane permeability in oocytes that causes striking cell toxicity. To minimise the toxicity in mammalian cells expressing CALHM6, either unknown regulatory mechanisms are present to ensure channel closure under normal conditions, or it is not usually localised in the plasma membrane. The latter is consistent with the dominant intracellular localisation of CALHM6 in BMDM and N2a cells and with the lack of response of CALHM6-expressing N2a cells to a $Ca^{2+}$ "add-back" protocol that activates CALHM1 and CALHM1/3 channels. During BMDM activation, CALHM6 was observed to translocate from intracellular compartments to the immunological synapse. We suggest that it is inserted into the membrane there where it provides a highly localised, and likely transient, way for metabolites to permeate through its large pore and activate NK cells.

In conclusion, we provide novel insights into macrophage-NK cell immune synapse and into the role of CALHM6 as a pro-inflammatory protein important for the timely activation of the innate immune response. We believe that future studies will find the role of CALHM6 not limited to *L. monocytogenes* infection response or anti-tumour DC therapy (Kasamatsu *et al*, 2014), but more broadly applicable to immune challenges in which the timing of the activation of immune cells and IFN-γ production by NK cells are important for the correct downstream function of the immune response. More broadly we provide novel insights into the physiological role of CALHMs as channel-forming proteins and characterise their only immune-specific member, CALHM6.

# Materials and Methods

## Reagents and Tools table

| Reagent or resource | Source | Identifier |
|---|---|---|
| **Antibodies and reagent for immunoblotting** | | |
| 4–20% miniprotean tgx precast gels 12 well | BioRad | 4568095 |
| 4–20% miniprotean tgx precast gels 15 well | BioRad | 4568096 |
| 4× Laemmli Sample Buffer, 1610747 | BioRad | 1610747 |
| Anti-HA Tag antibody | Merck | 05-904 RRID:AB_417380 |
| cOmplete EDTA-free protease inhibitor tablets | Sigma-Aldrich | 11873580001 |
| DTT | Sigma-Aldrich | GE17-1318-02 |
| Peroxidase AffiniPure Goat Anti-Rabbit IgG | Jackson Immuno/Stratech Scientific | 111-035-144-JIR RRID:AB_2307391 |

**Reagents and Tools table**   (continued)

| Reagent or resource | Source | Identifier |
| --- | --- | --- |
| Goat anti-mouse IgG1-hrp | SouthernBiotech | 1071-05 |
| Mouse Anti-FAM26F Antibody (G-12) | Santa Cruz | sc-515780 |
| Nativepage novex 4–16% bis-tris protein gels, 1.0 | BioRad | BN1002BOX |
| Nativepage running buffer kit | Thermo Fisher Scientific | BN2007 |
| Nativepage sample prep kit | Thermo Fisher Scientific | BN2008 |
| Rabbit GAPDH antibody | CST | 2118 RRID:AB_561053 |
| **Flow cytometry antibodies and reagents** | | |
| 16% Formaldehyde (w/v), Methanol-free-10× | Thermo Fisher Scientific | 28908 |
| APC anti-Mouse CD215 (Il-15rα) | Biolegend | 153505 RRID:AB_2734220 |
| APC anti-mouse NK-1.1 antibody | Biolegend | 108710 RRID:AB_313397 |
| BD cytofix/cytoperm | BD | 554722 |
| Brefeldin 1000× Protein Secretion Inhibitor | Biolegend | 420601 |
| CD11b anti-mouse BV711 | Biolegend | 101242 RRID:AB_2563310 |
| CD11b anti-mouse BV785 | Biolegend | 101243 RRID:AB_2561373 |
| CD11c anti-mouse PE | Biolegend | 117307 RRID:AB_313776 |
| CD19 anti-mouse APC-Cy7 | Biolegend | 115530 RRID:AB_830707 |
| CD27 anti-mouse PE/Dazzle | Biolegend | 124227 RRID:AB_2565793 |
| CD3 anti-mouse PE-Cy7 | Biolegend | 100220 RRID:AB_1732057 |
| CD4 anti-mouse PE/Dazzle | Biolegend | 100566 RRID:AB_2563685 |
| CD8a anti-mouse BV605 | Biolegend | 100743 RRID:AB_2561352 |
| DNAse I, grade ii | Sigma-Aldrich | 10104159001 |
| F4/80 anti-mouse FITC | Biolegend | 123108 RRID:AB_893502 |
| Granzyme B anti-mouse FITC | Biolegend | 515403 RRID:AB_2114575 |
| IFN-g anti-mouse PE | Biolegend | 505807 RRID:AB_315401 |
| Liberase | Sigma-Aldrich | 5401135001 |
| LIVE/DEAD® Fixable Near-IR Dead Cell Stain Kit, for 633 or 635 nm excitation | Thermo Fisher Scientific | L10119 |
| Ly6C anti-mouse BV605 | Biolegend | 127639 RRID:AB_2565880 |
| Ly-6c anti-mouse PE/Dazzle | Biolegend | 128044 RRID:AB_2566577 |
| P2X7R anti-mouse PE | BioLegend | 148703 RRID:AB_2650951 |
| Perm wash | BD | 554723 |
| Purified rat anti-mouse CD16/CD32 | BD | 553142 RRID:AB_394657 |
| Ultracomp ebeads™ compensation beads | eBioscience | 01-2222-42 |

**Reagents and Tools table**   (continued)

| Reagent or resource | Source | Identifier |
|---|---|---|
| **Microscopy antibodies and reagents** | | |
| Actin Red 555 | Thermo Fisher Scientific | R37112 |
| Alexa Fluor® 647 Goat anti-mouse IgG (minimal x-reactivity) Antibody | Biolegend | 405322 RRID:AB_2563045 |
| Diluent c for general membrane labelling | Thomas Scientific | C914J95 |
| Latex beads, polystyrene | Sigma-Aldrich | LB30-1ML |
| Mouse Anti-FAM26F Antibody (G-12) Alexa Fluor® 647 | Santa Cruz | sc-515780 AF647 |
| Mouse Anti-Flag Antibody, (M2) | Sigma-Aldrich | F3165 RRID:AB_439685 |
| PKH26 red fluorescent cell linker mini kit for general cell membrane labelling | Sigma-Aldrich | MINI26-1KT |
| Alexa Fluor 647 anti-DYKDDDDK Tag Antibody | Biolegend | 637315 RRID:AB_2716154 |
| Prolong® diamond antifade mounting with DAPI | Thermo Fisher Scientific | P36971 |
| Sheep blood in alsevers solution liquid oxoid | Thermo Fisher Scientific | 12977755 |
| **Chemicals, peptides, and recombinant proteins** | | |
| Recombinant M-CSF | Immunotools | 11343118 |
| Ampicillin | Sigma-Aldrich | A9518 |
| *E. coli* LPS 055:B5 | Sigma-Aldrich | L2637 |
| ARL67156 trisodium salt ATPase | Bio-Techne (R&D Systems) | 1283/10 |
| Cell Activation Cocktail (Without Brefeldin A) 500× | Biolegend | 423302 |
| Poly(I:C) | Sigma-Aldrich | P1530-25mg; CAS:42424-50-0 |
| Gentamicin | Sigma-Aldrich | G1397-10ML |
| Ionomycin, calcium salt | Thermo Fisher Scientific | I24222 |
| Nigericin | Sigma-Aldrich | N7143-5MG; CAS:28643-80-3 |
| Pen/strep glutamine | Thermo Fisher Scientific | 10378016 |
| Puromycin dihydrochloride | PeproTech | P8833 |
| Recombinant mouse ifn-γ (carrier-free) | Biolegend | 575304 |
| Recombinant mouse il-10 (carrier-free) | Biolegend | 575804 |
| Recombinant mouse il-12 (p70) (carrier-free) | Biolegend | 577004 |
| Recombinant mouse il-18 (carrier-free) | Biolegend | 592102 |
| HEPES (for transfection mix for retroviral transduction) | Thermo Fisher Scientific | 15630056 |
| Polybrene (for transfection mix for retroviral transduction) | Merck | TR-1003-G; CAS:28728-55-4 |
| DSS (Disuccinimidyl suberate) Crosslinking reagent | Thermo Fisher Scientific | 21655 |
| ATP | Thermo Fisher Scientific | 18330019 |
| Recombinant mouse il-4 (carrier-free) | Biolegend | 574304 |
| Recombinant mouse TNF (carrier-free) | Biolegend | 575204 |
| Recombinant mouse TGF-β1 (carrier-free) | Biolegend | 763104 |
| Recombinant murine GM-CSF | PeproTech | 250-05 |
| Zymosan a from saccharomyces cerevisiae | Sigma-Aldrich | Z4250 |
| Fura-2 AM | Invitrogen | F1225 |
| **Enzyme and reagents for transfection and transduction** | | |
| BamHI-HF | NEB | R3136S |
| Lipofectamine 2000 | Thermo Fisher Scientific | 11668019 |
| Lipofectamine 3000 | Thermo Fisher Scientific | L3000008 |
| Neb 5-alpha competent *E. coli* (high efficiency) | NEB | C2987H |

**Reagents and Tools table**  (continued)

| Reagent or resource | Source | Identifier |
|---|---|---|
| Phusion polymerase | NEB | M0530S |
| T4 dna ligase | NEB | M0202L |
| Polybrene | Merck | TR-1003-G |
| NotI HF | NEB | R3189S |
| Smart MMLV reverse transcriptase 20,000 | Takara Bio Europe SAS | 639524 |
| Collagenase type 2 | Worthington-Biochem | CLS-2 |
| AgeI-HF | NEB | R3552S |
| HindIII-HF | NEB | R3104S |
| SalI-HF | NEB | R3138S |
| **Assays and kits** | | |
| IL-1 beta mouse uncoated ELISA Kit | Thermo Fisher Scientific | 88-7013-22 |
| Cytotox 96® non-radioactive cytotoxicity assay (ldh assay) | Promega | G1780 |
| Enliten rluciferase/luciferin reagent | Promega | FF2021 |
| Magnisort™ mouse CD49b positive selection kit | eBioscience | 8802-6862-74 |
| Magnisort™ mouse F4/80 positive selection kit | eBioscience | 8802-6863-74 |
| Mouse IFNg elisa | eBioscience | 88-7314-77 |
| Mouse IL-18 elisa | eBioscience | BMS618-3TEN |
| pHrodo Red Dextran, 10,000 MW, for Endocytosis | Thermo Fisher Scientific | P10361 |
| pHrodo Red Zymosan Bioparticles Conjugate for Phagocytosis | Thermo Fisher Scientific | P35364 |
| Qiagen plasmid plus maxi kit | Qiagen | 12963 |
| NEBNext rRNA Depletion Kit (Human/Mouse/Rat) | NEB | #E7400 |
| NEBNext Ultra™ II Directional RNA Library Prep Kit for Illumina | NEB | #E7765 |
| Anti-F4/89 MicroBeads UltraPure, mouse | Miltenyi Biotec | 130-110-443 |
| MS columns | Miltenyi Biotec | 130-042-201 |
| LEGENDplex Mouse inflammation panel | BioLegend | 740150 |
| ATP Bioluminescent Assay Kit | Sigma | FLAA-1KT |
| Griess Reagent System | Promega | G2930 |
| PowerUp SYBR Green Master Mix | Thermo Fisher Scientific | A25777 |
| Qiaquick gel extraction kit | Qiagen | 28704 |
| Rneasy mini kit | Qiagen | 74106 |
| Maxi Prep Kit | Qiagen | 12123 |
| RNA kit-mMessage mMachine SP6 | Invitrogen | AM1340 |
| **Experimental models: cell lines** | | |
| HEK293T cells | Sigma Aldrich | 12022001-1VL |
| N2a (Neuro-2a) cells | ATCC | CCL-131 |
| Platinum-E (PlatE) retroviral packaging cell line | eBioscience | RV-101 |
| S-S.1 (TIB-111) | American Type Culture Collection | TIB-111 |
| *Xenopus* oocytes from *Xenopus* Laevis | Xenopus 1Corp | 4121 |
| **Experimental models: organisms/strains** | | |
| C57BL/6J Flp mice (ACTB:FLPe B6J) | The Jackson Laboratory | #005703 |
| C57BL/6J Itgax$^{cre/-}$ mice (B6.Cg-Tg(Itgax-cre)1-1Reiz/J) | The Jackson Laboratory | #008068 |
| C57BL/6J PGK$^{Cre/-}$ mice | The Jackson Laboratory | #020811 |
| C57BL/6J Rag$^{-/-}$ mice (B6.129S7-Rag1$^{tm1Mom}$/J) | The Jackson Laboratory | #002216 |

**Reagents and Tools table**   (continued)

| Reagent or resource | Source | Identifier |
|---|---|---|
| C57BL/6N FAM26Ffl/fl mice | From EUCOMM repository | Generated by Jelena Bezbradica in the Medzhitov lab (Yale School of Medicine, New Haven, CT) |
| **Plasmids** | | |
| pMIGRMCS retroviral packaging plasmid expressing GFP, CALHM6-FLAG, CALHM6-HA, CALHM1-HA | Plasmid backbone from Prof Kate Schroder lab (University of Queensland, AU) | Sequence available upon request |
| pMMLV mammalian gene expression plasmid expressing IRES eGFP and CALHM6-eGFP | VectorBuilder | Sequence available upon request |
| mCALHM6 untagged plasmid | Origene Technologies Inc | MR204443 |
| WT-mCALHM6 and E119R-mCALHM6 WT-mCALHM6-GFP and E11R-mCALHM6-GFP in pBF vector | Subclone with SalI and AgeI sites into pBF vector | Sequence available upon request |
| mCALHM6-GFP in pCMV6-A-BSD vector | Subclone with BamHI and HindIII sites into pCMV6-A-BSD vector | Sequence available upon request |
| hCALHM1 and mCALHM6 in pIRES2-AcGFP1 vector | Subclone with SalI and BamHI sites into pIRES2-AcGFP1 vector | Sequence available upon request |
| **Software and algorithms** | | |
| GraphPad Prism | http://www.graphpad.com/ | RRID: SCR_002798 |
| BioRender | http://biorender.com | RRID:SCR_018361 |
| SRplot | http://www.bioinformatics.com.cn | |
| Fiji | https://fiji.sc/ | RRID: SCR_002285 |
| FlowJo | https://fiji.sc/ | RRID: SCR_008520 |
| Pulse 8.80 | http://heka.com/ | Pulse 8.80 version |
| Igor program | https://www.wavemetrics.com/ | Igor 8.0 version |
| **Primers for qPCR** | | |

| Gene | Forward | Reverse |
|---|---|---|
| Calhm1 | TGAGATCTATGATGGGAACTG | CAATGTCAATGTAGTGGGAC |
| Calhm2 | AGTATGAGTCTCAGCTCTTC | AGTAATGTTTGAGGCACTTG |
| Calhm3 | TGATGGTGGAGGAGTGGC | GCACACACAAAACACTTGCC |
| Calhm4 | CACAGATACCAGTCACAAATG | CGATGTCTAATAAGAAGTTGGG |
| Calhm5 | GATAACACAGTGGGAAACTG | CTTACAAATCGTGGCAGACC |
| Calhm6 | CTCTTTCTACCAATGTGCTG | CTATCAGAATCCAACCGAAC |
| Hprt | GTTGGATACAGGCCAGACTTTGTTG | CCAGTTTCACTAATGACACAAACG |
| Ifna14 | TCTCTCTCAGGTAAACAGTG | TAGACTCCTTCTGCAATGAC |
| Ifna4 | CAAAATCCTTCCTGTCCTTC | ATGATAGAGCTACTACTGGTC |
| Ifnb1 | AACTCTGTTTTCCTTTGACC | AACTTCCAAAACTGAAGACC |
| Il10 | CAGGACTTTAAGGGTTACTTG | ATTTTCACAGGGGAGAAATC |
| Il1r2 | GAAGAGACTTCTTTGACTGTG | AAAACTATGTGGAAGTGTCG |
| Il12a | GAAGACATCGATCGATCATGAAGAC | CTCTTGTTGTGGAAGAATC |
| Il12b | CATCAGGGACATCATCAAAC | CTCTGTCTCCTTCATCTTTTC |
| **Primers for genotyping** | | |
| Calhm6 primer 1 | GTGTAGCAAAGCCTCAGGACAGGT | |
| Calhm6 primer 2 | CCAACTGACCTTGGGCAAGAACAT | |
| Calhm6 primer 3 | CACACCTCCCCCTGAACCTGAAA | |
| Calhm6 primer 7 | CTTTCTGCCCAATCCTCCTG | |
| Calhm6 primer 8 | GTTTCCCTAACTGGGCTGTC | |
| Cd11c-control | CAAATGTTGCTTGTCTGGTG | GTCAGTCGAGTGCACAGTTT |
| Cd11c-Cre | ACTTGGCAGCTGTCTCCAAG | GCGAACATCTTCAGGTTCTG |

Reagents and Tools table (continued)

| Reagent or resource | Source | Identifier |
| --- | --- | --- |
| Flp | ATAGCAGCTTTGCTCCTTCG | TGGCTCATCACCTTCCTCTT |
| Flp-Control | CTAGGCCACAGAATTGAAAGATCT | GTAGGTGGAAATCTAGCATCATCC |
| LacZ cassette | GGTAAACTGGCTCGGATTAGGG | TTGACTGTAGCGGCTGATGTTG |
| Pgk-Cre | CGTTTTCTGAGCATACCTGGA | ATTCTCCCACCGTCAGTACG |
| Rag | TGGATGTGGAATGTGTGCGAG | CATTCCATCGCAAGACTCCT |
| Rag-control | TCTGGACTTGCCTCCTCTGT | CATTCCATCGCAAGACTCCT |

## Methods and Protocols

### In vitro stimulation of BMDM, BMDC and NK cells

Bone marrow-derived macrophages and BMDCs, were generated by culturing frozen BM cells for 7 days in RPMI 1640 complete medium (10% low endotoxin heat inactivated foetal bovine serum (FBS), 100 units/ml penicillin, 100 μg/ml streptomycin and 5.84 μg/ml glutamine, 10 μM HEPES) with respectively 50 ng/ml recombinant human M-CSF (rhM-CSF) or 10 ng/ml mouse GM-CSF. Unless specifically stated, for in vitro stimulation we used ultra-pure LPS (100 ng/ml), ultra-pure Zymosan (100 μg/ml), ultra-pure Poly(I:C) (50 μg/ml), IFN-γ (10 ng/ml), TGF-β (10 ng/ml), IL-10 (10 ng/ml) and IL-4 (10 ng/ml), for the duration of the stimulation see figures and figure legends. For details see material source table. For in vitro L. monocytogenes stimulation, frozen, stationary phase, OVA expressing L. monocytogenes (kindly donated by Dr Gerard's lab, Kennedy Institute of Rheumatology, University of Oxford, originally described in Pope et al, 2001) were used at a multiplicity of infection (MOI) of 1. BMDM were infected for 1 h before being washed, and supernatant supplemented with Gentamicin (5 μg/ml), for 6 h in total. NK cells in vitro stimulation was performed on FACS sorted splenocytes (CD3$^-$NK1.1$^+$) stimulated with PMA/Ionomycin 1:500 (Phorbol 12-Myristate 13-Acetate (40.5 μM), Ionomycin (670 μM), IL-12 (50 ng/ml), TNF (50 ng/ml) and or IL-18 (50 ng/ml) for 6 h).

### Quantitative real-time PCR

RNA was extracted from BMDM, F4/80$^+$ cells or CD11c$^+$ cells isolated using MACS Microbeads according to manufacturer instructions. For RNA extraction we used RNeasy Mini kit. Complementary DNA was obtained by using 1 μg of starting RNA where possible and for the reverse transcriptase reaction oligo-dT and SMART MMLV Reverse Transcriptase were used. Quantitative real-time PCR was performed using SYB Green I Nucleic Acid Gel Stain according to manufacturer instructions. mRNA abundance was normalised to Hprt expression.

### Immunoblots

For Native BluePAGE gel, BMDM were lysed in 50 μl fridge cold 1× NativePAGE Sample Buffer containing 1% Digitonin or 1% dichlorodiphenyltrichloroethane (DDT) in the presence of cOmplete Protease Inhibitor Cocktail. NativePAGE 5% G-250 sample additive was added to the samples that were run on NativePAGE Bis-Tris gels in NativePAGE dark blue running buffer. For SDS–PAGE gels cells were lysed in sodium dodecyl sulphate (SDS) lysis buffer (2% SDS 66 mM, Tris pH 7.4) or, for non-denaturing lysis conditions, in Triton buffer (50 mM Tris pH 7.4, 150 mM NaCl, 1% Triton X-100 with fresh cOmplete Protease Inhibitor Cocktail). Lysates were mixed with 4× Laemmli sample buffer containing 20 mM DTT, unless otherwise specified, and run on an SDS–PAGE gels 4–20%.

### Mouse generation

C57BL/6N Calhm6$^{fl/fl}$ mice were generated by Jelena S Bezbradica, in Ruslan Medzhitov laboratory (Yale School of Medicine, New Haven, CT) using pre-targeted Fam26f$^{tm2a(EUCOMM)Wtsi}$ embryonic stem cells (reporter-tagged insertion with conditional potential) by EUCOMM on C57BL/6N background (Friedel et al, 2007). To generate compete Calhm6$^{-/-}$ mice, we crossed the original C57BL/6N Fam26f$^{tm2a(EUCOMM)Wtsi}$ mice with C57BL/6J Tg(Pgk1-cre) line (generously donated by Tal Arnon's lab, Kennedy Institute of Rheumatology, Oxford, UK, original Tg(Pgk1-cre)1Lni strain from Jackson laboratories). Tg(Pgk1-cre) allele directs the expression of Cre recombinase under the control of a phosphoglycerate kinase promoter and deletes genes in all tissues at the diploid phase of oogenesis. The cross resulted in Calhm6 deletion in all tissues (allele B6.FAM26F$^{tm2b}$). We then bred out Tg(Pgk1-cre) after Calhm6 gene deletion; the subsequent Calhm6$^{-/-}$ line did not carry the Pgk-cre transgene. From the same crossing, wild type controls, on the same C57BL/6JN background were generated.

To generate CD11c-conditional KO mice we crossed our original C57BL/6N Calhm6$^{fl/fl}$ mice with a flippase-expressing C57BL/6J strain to excise the LacZ-containing cassette that interferes with the expression of CALHM6. The mice were then crossed with an C57BL6/J Itgax$^{cre/-}$ expressing strain (B6.Cg-Tg(Itgax-cre)1-1Reiz/J Jackson laboratory) to obtain Itgax$^{cre/-}$Calhm6$^{fl/fl}$ mice.

We generated Rag1$^{-/-}$Calhm6$^{-/-}$ strain by crossing C57BL/6N Calhm6$^{-/-}$ mice with C57BL/6J Rag1$^{-/-}$ mice (B6.129S7-Rag1$^{tm1Mom}$/J Jackson laboratory).

### Listeria monocytogenes infection

Log phase L. monocytogenes (Pope et al, 2001) resuspended in PBS was injected i.p. in mice at a concentration of $3 \times 10^4$ or $3 \times 10^5$ CFU/mouse. For determination of CFU, spleens or livers were lysed by centrifugation with Precellys beads (1.4 mm) at 10,000 rpm on a Quiashredder for 10 min in 1 ml PBS. The lysate obtained was plated in several dilutions on BHI-Agar coated plates without antibiotics. CFU load per spleen was calculated based on the number of colonies, organ weight, and dilution factor. For the determination of IFN-γ producing cells, splenocytes from infected mice were cultured for an additional 4 h in RPMI 1640 complete medium with Brefeldin A (5 μg/ml) to block protein secretion before intracellular staining. For flow cytometry staining, cells were incubated with buffer containing 0.5% normal mouse serum and 1:200 Fc blocking

antibodies to avoid nonspecific staining. NK cells were identified as $CD3^-NK1.1^+$ cells.

### Griess assay

Bone marrow-derived macrophages were stimulated overnight with LPS (100 ng/ml) or Poly(I:C) (50 μg/ml). Fifty microliters of supernatant from the cell culture was mixed with sulphanilamide solution (1% sulphanilamide in 5% phosphoric acid) and incubated for 10 min at room temperature before 50 μl of N-1-naphthylethylenediamine dihydrochloride were added on top according to manufacturer instructions (Griess Reagent System, Promega) and absorbance measured.

### Co-culture experiments

For BMDM/BMDC-NK cells co-culture, NK cells were freshly isolated from spleens with MagniSort mouse NK cell Negative Selection Kit and 300,000 NK cells were co-cultured with 150,000 BMDM or BMDCs in RPMI 1640 complete medium for 24 h with or without Poly(I:C) (50 μg/ml) stimulation. For $F4/80^+$ and $CD3^-NK1.1^+$ cell co-culture, $F4/80^+$ cells were obtained from $Rag1^{-/-}$ and $Ra1g^{-/-}Calhm6^{-/-}$ mouse spleens by using the MagniSort Mouse F4/80 Positive selection kit. The splenocyte flow through was stained for flow cytometry to isolate NK cells ($CD3^-NK1.1^+$). 100,000 $F4/80^+$ cells were co-cultured with 200,000 $CD3^-NK1.1^+$ cells for 24 h with or without Poly(I:C) (50 μg/ml) stimulation. For whole spleen cultures, single cell suspension was obtained from $Rag1^{-/-}$ and $Rag1^{-/-}Calhm6^{-/-}$ spleens and 300,000 cells/well were plated with or without Poly(I:C) (50 μg/ml) stimulation for 24 h.

### HEK293T transfection and BMDM transduction

Mouse CALHM6-HA, CALHM6-FLAG and CALHM6-ΔNH-HA expressing plasmids were all generated by cloning mouse full-length *Calhm6*, or ΔNH-truncated *Calhm6* (aa 1–8) from cDNA isolated from BMDM into the pMIG retroviral plasmid backbones kindly donated by the Kate Schroder laboratory (The University of Queensland, Australia) containing either HA or FLAG tag at the C-terminus, and IRES-GFP as a reporter gene. Mouse CALHM6-eGFP (C-terminally tagged) plasmid and empty vector control were ordered from VectorBuilder and came on a pMMLV background. For HEK293 transfection we used Lipofectamine 2000. Transfection efficiency was based on %GFP$^+$ cells measured by flow cytometry. In order to generate the retrovirus necessary to transduce BMDM our plasmids were first transfected CALHM6-expressing plasmids into the Retroviral Packaging cell line Plat-E cells with Lipofectamine 2000. Fresh BM cells were collected 1 day before transfection, and cultured in 50 ng/ml rhM-CSF in RPMI 1640 complete medium for a day. We then performed two rounds of bone marrow transduction, on two consecutive days, using Plat-E supernatant containing retroviruses encoding *Calhm6*. The procedure is described in detail in (Fischer *et al*, 2021). Transduced BMDM were harvested on day 7.

### N2a cell culture and transfection

The N2a mouse neuroblastoma cell line was cultured in Eagle's minimum essential medium supplemented with 10% FBS and 0.5 × penicillin/streptomycin (Invitrogen) at 37°C in a humidified incubator with 5% $CO_2$ (Ma *et al*, 2012), N2a cells were transfected with human CALHM1 or mouse CALHM6 in pIRES2-AcGFP1 vector by Lipofectamine 3000 (Invitrogen) and used 24 h later.

### RNA sequencing

RNA purification from BMDM was done using RNeasy Mini Kit from Qiagen, following manufacturer's instructions. Agilent 2100 Bioanalyzer was used to assay RNA quality and integrity and the RNA was depleted of ribosomal RNA using NEB rRNA depletion kit. NEB Nest Ultra II Directional RNA library kit for Illumina was used for library preparation following manufacturers instruction. Briefly, rRNA depleted fraction was fragmented, converted to double-stranded cDNA and adapter-ligated following end-repair and dA-tailing. Adapter-ligated libraries were generated by PCR using Illumina PE primers and the resulting purified cDNA libraries were applied to an Illumina flow for cluster generation followed by sequencing on an Illumina instrument following manufacturer's instructions. The sequence data quality was checked using FastQC and MultiQC software. Raw sequence reads were processed to remove adapter sequences and low-quality bases using fastp. The QC passed reads were mapped onto indexed Mouse reference genome (GRCm 38.90) using STAR v2 aligner. The PCR and optical duplicates were marked and removed using Picard tools. Differential expression analysis was carried out using edgeR package after normalising the data based on trimmed mean of *M* values. Genes with absolute log2 fold change ≥ 2 and *P*-value ≤ 0.05 were considered significant. Enrichment analysis for Biological process, Molecular function, and Cellular component was performed using the Gene Ontology (GO) resource (Ashburner *et al*, 2000). GO and pathway terms with, multiple test adjusted *P*-value ≤ 0.05 are considered significant.

### Confocal imaging

All cell images were taken on an Olympus 980 microscope with a 60× magnification. Transduced BMDM expressing CALHM6-eGFP or empty vector control were grown overnight on a coverslip, 300,000 cells/well so as to adhere. Were indicated cells were treated with Poly(I:C) (50 μg/ml) and IFN-γ (10 ng/ml) for 6 h before imaging. To visualise IS, 300,000 freshly sorted splenic $CD3^-NK1.1^+$ cells or 3,000,000 polystyrene beads (3 μm size) were added on top of transduced BMDM before gentle centrifugation, and the plates were placed back in the incubator. After 0, 10, 20 or 40 min supernatant was removed and replaced directly with 4% PFA for 10 min at room temperature (RT) for fixation. Cells were permeabilised with a solution of 0.1% Triton X-100 in PBS for 20 min at RT. To avoid unspecific staining cells were blocked in blocking buffer (5% FCS in PBS, 0.2% NaN3, 1:200 Fc block and 1% mouse serum) for 30 min at RT and then stained with fluorescently conjugated phalloidin. To visualise BMDM-sRBC synapses, sRBC were opsonised for 1 h using supernatant from the TIB-111 cell line from the American Type Culture Collection, a B-lymphocyte hybridoma, kindly provided by the Medzhitov laboratory, containing anti-sRBC IgG2a antibody. sRBC were washed and labelled with PKH26 membrane dye, according to manufacturer instruction (but with 10× more cells than suggested). 3,000,000 sRBC were added to the BMDM culture and the staining carried out as before.

### Phagocytosis assays

For sRBC phagocytosis fresh sRBC were labelled with PKH26 and opsonised with anti-sRBC IgG2a or left untreated. sRBC were then washed and added into BMDM-containing wells at an sRBC/BMDM ratio of 20:1. The cells were incubated at 37 (C) and after 5, 15 or 45 min washed with 4 (C) PBS, lifted with Trypsin-EDTA, and fixed

in 4% PFA before flow cytometry run. Alternatively, pHrodo Red Zymosan (25 μg/ml) or pHrodo Red Dextran 10,000 MW (50 μg/ml) were added to the BMDM for the time indicated before analysis by flow cytometry.

### Ca²⁺ dependent ATP release assay

For ATP release BMDM, transduced BMDM, transfected HEK293T cells or Plat-E cells were plated with a density of 100,000 cells/well in a 96-well plate. BMDM were stimulated with IFN-γ (10 ng/ml) for 6 h or left untreated. To measure ATP release in low extracellular calcium conditions RPMI 1640 complete medium was replaced with 100 μl of 37 (C) low Ca²⁺ Isotonic Buffer (150 mM NaCl, 5 mM KCl, 10 mM HEPES, 10 mM glucose, 2.5 mM EDTA, pH adjusted to 7.4) for 20 min with ATPase inhibitor ARL67156 (1 mM) added to the isotonic buffer. Final ATP concentration was measured by ENLITEN ATP assay system, with 10 μl of supernatant added to 90 μl ENLITEN buffer before luciferase activity measurement. For other ATP release conditions RPMI was replaced with Excess K⁺ buffer (117.5 mM KCl, 37.5 mM NaCl, 10 mM HEPES, 10 mM Glucose) or Isotonic buffer with calcium (140 mM NaCl, 5 mM CaCl, 10 mM HEPES, 10 mM Glucose) and Ionomycin (25 μM). All buffers contained the ATPase inhibitor ARL67156.

### Macrophage culture and stimulation for steady state ATP release

Macrophages were cultured for 7 days, and then macrophages were plated on poly-L-lysine-coated 12-mm diameter coverslips and used 48 h later. Macrophages were stimulated for 3 h with Poly(I:C) (50 μg/ml) and LPS (100 ng/ml) in the presence of the ATPase inhibitor ARL67156 (1 mM). Control macrophages were treated with 1 mM ARL67156 only. ATP release from macrophages was measured as we described previously for taste bud and N2a cells (Taruno *et al*, 2013). One hundred microliter medium was collected from each well and added to an equal volume of ATP assay solution FLAA-1KT (Sigma-Aldrich) to determine ATP concentration by the luciferin-luciferase assay with a microplate luminometer (Synergy 2, BioTek). The bath solution contained (in mM) 140 NaCl, 5 KCl, 2 CaCl₂, 1 MgCl₂, 10 glucose, 5 Na-pyruvate and 10 HEPES, pH 7.4 with NaOH.

### Single-cell calcium imaging

Macrophages and N2a cells were plated on poly-L-lysine–coated glass coverslips. Cells were loaded with Fura-2 by incubation in medium containing 4 μM Fura-2-AM for 30 min or 2 μM for 45 min for macrophages and N2a cells, respectively, followed by 5 min continuous perfusion with dye-free 2-mM Ca²⁺ bath solution. The background-subtracted emitted light at 520 nm upon excitation with 340 and 380 nm light was used to calculate the F340/380 ratio with VisiView.

2 mM Ca²⁺ bath solution (in mM): 140 NaCl, 5 KCl, 2 CaCl₂, 10 Glucose, 5 Na-Pyruvate, 10 HEPES, pH 7.4 with NaOH.

0-Ca²⁺ bath solution (in mM): 140 NaCl, 5 KCl, 5 EGTA, 10 Glucose, 5 Na-Pyruvate, 10 HEPES, pH 7.4 with NaOH.

### Inflammasome activation

Bone marrow-derived macrophages were plated at a density of 100,000 cell/well the day before the experiment. On the day of the experiment cells were treated with Poly(I:C) (50 μg/ml) or LPS (100 ng/ml) for 1 h or 4 h, before adding nigericin (10 μg/ml) or

ATP (2.5 mM). After 1 h cell-free supernatant was collected and used to assess the level of pyroptotic cell death by performing an LDH assay. 100% cell death for normalisation was obtained by lysing cells with 0.1% Triton X-100.

### Statistical analysis

Statistical analysis was performed with GraphPad Prism 9 unless otherwise indicated. One-way ANOVA with multiple comparisons was used to determine differences between 3 or more unrelated groups (e.g. *Calhm6* expression in different stimulations), two-way ANOVA was used when considering changes in a variable according to the levels of two categorical variables (e.g. weight according to day and genotype), Mann–Whitney test was used to compare differences in a variable for two independent groups when the data are not normally distributed (e.g. CFU/g of spleen between genotypes) and Kruskal–Wallis test with multiple comparisons was used when we had three or more categorical independent groups non normally distributed (e.g. cytokine release according to treatment and genotype). In the figures significant differences are only marked when detected between different genotypes in the same group or between positive and negative control. Non-significant differences are not indicated for ease of reading.

## Data availability

GEO accession for the RNA sequencing GSE223008 ([https://www.ncbi.nlm.nih.gov/geo/query/acc.cgi?acc=GSE223008](https://www.ncbi.nlm.nih.gov/geo/query/acc.cgi?acc=GSE223008)).

**Expanded View** for this article is available online.

## Acknowledgements

We acknowledge Prof. Ruslan Medzhitov for his support in the generation of *Calhm6*⁻/⁻ mice and helpful feedback. We would like to thank Dr Audrey Gerard for kindly providing us with *L. monocytogenes* and prof. Kate Schroder for the pMIG plasmid backbone. We also acknowledge Lenka Milojevic for her help with plasmid generation. The graphical model was created with Biorender.com. RNAseq analysis was performed by Clevergene Biocorp Pvt. Ltd. (Bengaluru, India). Animal care was in accordance with institutional guidelines. SD and FAF are supported by Kennedy Trust, KTPS Studentships. BD is supported by EMBO postdoctoral fellowship (ALTF 108-2021). JSB is supported by the Kennedy Trust, KTRR start-up fellowship (KENN 15 16 06), MRC grants (MR/W001217/1, MR/S000623/1), MRF-150-0001-STD-MIRK-C0809, and John Fell Oxford University Press Research Fund (0010732). SJ is a Research Career Scientist of the Department of Veterans Affairs (IK6 BX004595), and supported by VA Merit Award (IO1 BX001444) and NIH RO1 grant (AI137082). ZM and JKF are supported by NIH/NIDCD RO1DC08278.

## Author contributions

**Sara Danielli:** Formal analysis; investigation; writing – original draft. **Zhongming Ma:** Formal analysis; investigation. **Eirini Pantazi:** Investigation. **Amrendra Kumar:** Investigation. **Benjamin Demarco:** Investigation. **Fabian A Fischer:** Investigation. **Usha Paudel:** Formal analysis; investigation. **Jillian Weissenrieder:** Formal analysis; investigation. **Robert J Lee:** Formal analysis; investigation. **Sebastian Joyce:** Conceptualization. **J Kevin Foskett:** Conceptualization; formal analysis; supervision; funding acquisition; investigation; writing – review and editing. **Jelena S Bezbradica:**

Conceptualization; formal analysis; supervision; funding acquisition; investigation; writing – review and editing.

## Disclosure and competing interests statement

The authors declare that they have no conflict of interest.

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
