## [Review Process File · The EMBO Journal]

The ion channel CALHM6 controls bacterial infection-induced cellular cross-talk at the immunological synapse

Sara Danielli, Zhongming Ma, Eirini Pantazi, Amrendra Kumar, Benjamin Demarco, Fabian Fischer, Usha Paudel, Jillian Weissenrieder, Robert Lee, Sebastian Joyce, J. Kevin Foskett, and Jelena Bezbradica

DOI: [10.15252/emboj.2022111450](https://doi.org/10.15252/emboj.2022111450)

Corresponding author(s): Jelena Bezbradica (jelena.bezbradica@kennedy.ox.ac.uk) , J. Kevin Foskett (foskett@pennmedicine.upenn.edu)

Review Timeline:

Submission Date:	15th Apr 22
Editorial Decision:	25th May 22
Revision Received:	7th Nov 22
Editorial Decision:	21st Dec 22
Revision Received:	19th Jan 23
Accepted:	26th Jan 23

Editor: Karin Dumstrei

Transaction Report:

Dear Jelena,

Thank you for submitting your manuscript to The EMBO Journal. Your study has now been seen by three referees and their comments are provided below.

As you can see from the comments, the referees find the analysis interesting. However, they also all raise the issue that we would need some further mechanistic insight for consideration here. Referee #3 suggests to look at Calcium homeostasis as a possible mechanism. Should you be able to add more insight into how CALHM6 contributes to macrophage-NK cell communication then I would like to invite you to submit a revised version. We don't need the full mechanism but some is needed.

It would be helpful to discuss the revisions further via email or a video call. Let me know when works best for you.

Thank you for the opportunity to consider your work for publication. I look forward to discussing the revisions further with you
with best wishes

Karin

Karin Dumstrei, PhD
Senior Editor
The EMBO Journal

I have attached a PDF with helpful tips on how to prepare the revised version.

Guide For Authors: <https://www.embopress.org/page/journal/14602075/authorguide>

We realize that it is difficult to revise to a specific deadline. In the interest of protecting the conceptual advance provided by the work, we recommend a revision within 3 months (23rd Aug 2022). As a matter of policy, competing manuscripts published during this period will not negatively impact on our assessment of the conceptual advance presented by your study. However, we request that you contact the editor as soon as possible upon publication of any related work, to discuss how to proceed.

Please discuss the revision progress ahead of this time with the editor if you require more time to complete the revisions.

Use the link below to submit your revision:

Referee #1:

In this MS the authors present a nice body of data to support that the membrane ion channel CALHM6 is an immune synaptic channel-forming protein that contributes to neurotransmitter-like signal exchange between myeloid and NK cells. The role of this protein seems to be restricted to the macrophage - NK cell synapse. The findings are both interesting and potentially important.

Major issues

1. The abstract says "they fail to control *L. monocytogenes* burden at the peak of infection". The results show CALHM6 is important for regulating the early innate control of *L. monocytogenes* infection. This is a bit misleading- it regulates the bacterial burden, but does not control it as the bacterial levels have dropped by D7 (which is interesting as what else kicks in to help regulate IFN γ production?). Presumably the IFN γ is also required for the development of clearing adaptive immunity against this pathogen as well? IL18 normally precedes the induction of IFN γ and the authors measure this later on, but some discussion a bit earlier in the MS might help to strengthen the arguments here. The IL18 data only plays a supporting role

to IL1 in the MS at the moment, but it could be a reason why the differences in early/late NK cell IFN γ production is seen. Maybe the authors may need to reorder some of the data in the MS to address this? The data showing this may be more appropriate for earlier in the MS? Reword to say CALHM6 is important for regulating the early innate control of *L. monocytogenes* infection. Simple clarification of the various sources of IFN γ and its relationship to the different cell types involved in innate and adaptive immunity would be useful here to support the authors arguments.

2. "Cytokine response was then compensated, albeit with a delayed kinetics, to finally allow full bacterial clearance by day 7". This is a bit unclear: do the authors mean a hypothesis whereby NK cell IFN γ was reduced on D1, but IFN γ production from other cells compensated for this such that by D3 it was elevated? If so can the authors add to the text an explanation as to how and why this might happen.

3. "so the early defect in NK cell derived IFN-g would explain uncontrolled proliferation of the bacteria" better here to say enhanced proliferation of the bacteria as there is no evidence to show the bacterial growth is uncontrolled: this would require a number of different experiments in different KO mouse strains which is beyond the scope of this MS

4. The images in Fig 5 D-F are too small and it is difficult to see what is happening. Can the authors get better images supporting CALHM6 localisation to the NK immune synapse?

5. Do the authors have any hypothesis/explanation as to why CALHM6 would only be found in the macrophage - NK cell synapse? One might have expected a IS regulatory protein like this to be present in all the antigen presenting cells that form ISs? Does another CALHM perhaps perform similar functions in DCs?

Minor issues

Title: What does synaptic channel mean? Consider changing the title to immune synaptic channel protein for clarity

Abstract

"the timing of innate immune response" to: the timing of innate immune responses

Referee #2:

The authors investigate the role of the CALMH6 protein (also known as INAM), a voltage-gated oligomeric ATP channel, for NK cell activation and NK-myeloid cell crosstalk. In extension of previous data, they show that CALMH6 is the only molecule of this family which is inducibly expressed by PAMP-activated myeloid cells. A newly generated *Calmh6^{fl/fl}* mouse line shows that control of *Listeria monocytogenes* is impaired at day 3 but bacterial clearance by day 7 is not affected. While *Calmh6^{-/-}* NK cells were impaired in their propensity to produce IFN γ early after stimulation (polyIC or *L. monocytogenes*) they produced normal levels of IFN γ at later timepoints. Cell type-specific deletion of *Calmh6* in dendritic cells did not lead to any differences in NK cell granzyme B and IFN γ expression. Genome-wide transcriptional analysis of bone marrow-derived macrophages of *Calmh6^{-/-}* and control mice revealed only few differentially expressed genes, many of which were part of synaptic pathways. In particular genes associated with vesicle trafficking and fusion, or formation of synaptic contacts were reduced in *Calmh6^{-/-}* macrophages. Biochemical and CryoEM studies show that CALMH6 forms an oligomer and may act as a channel but the signals opening the CALMH6 channel and thereby affecting NK cell function remain unknown.

General concerns

The paper addresses an interesting topic, how CALMH6 enhances NK cell activation possibly by amplifying NK cell-macrophage crosstalk. The presented data is of good quality but, overall, new insights are limiting. Much of the data confirms previous work in another disease model (refs. 4, 5). A newly generated tissue-specific allele of the *Calmh6* gene is a valuable tool but its potential is not fully explored. The most interesting experiments are in Figures 5 and 6 but mainly negative data are presented. From the data presented, it remains unclear how CALMH6 amplifies NK cell function. More mechanistic insights would be expected from a paper in *EMBO J*.

The following specific concerns are raised.

1. The paper does not provide a mechanistic understanding of how CALMH6 contributes to NK cell activation. Without such data the significance is rather limiting. Further exploration of CALMH6 in the immunological synapse or its function as a channel would be needed.

2. Previous data suggested that NK cells can also express CALMH6? How does NK cell function look in mice with a deletion of *Calmh6* gene in NK cells (*Ncr1-Cre*).

3. The authors present data from mice lacking CALMH6 in dendritic cells (using *CD11c-Cre*) which did not change NK cell function. This should be further explored by using deletion of CALMH6 in macrophages, pDC, NK cells or all immune cells (*Vav-Cre*).

4. The establishment of a co-culture system of NK cells and myeloid (or other accessory cells) that would allow to assess how CALMH6 enhances NK cell function would be extremely helpful to gain additional mechanistic insights.

5. The authors claim that the production of type I IFNs is not altered in *Calhm6*^{-/-} mice. However, this is based on the assessment of type I IFN response genes (Figure S4). Direct measurements of serum type I IFNs should be provided.

Referee #3:

CALHM channels are ion channels with some features reminiscent of VRAC, pannexins, connexins. They are believed to conduct cations (with poor discrimination), anions and ATP. Potential functions of CAHLM6 in the immune systems are not well understood. The present ms by Danielli et al analyzes CALHM6 (Calcium homeostasis modulator ion channels) function in macrophages.

Pros

The authors convincingly show that CAHLM6 is the only CAHLM channel expressed in murine macrophages upon pathogen-dependent stimulation. They clearly show that CAHLM6 expression is upregulated by proinflammatory signals and downregulated by anti-inflammatory ones. The authors furthermore report that CAHLM6^{-/-} animals control infection by the bacteria *Listeria monocytogenes* not as well as wt mice with IFN- γ production by NK cells being delayed/decreased. Immune development is, however, normal. Finally, the authors try to show that CAHLM6 is segregated to the immune synapse between macrophages and NK cells, which kind of links the macrophage CAHLM6 expression with the NK problems in maintaining high INF- γ levels. With respect to immune synapse segregation of CAHLM6 I see some problems (see below).

Cons

The authors cannot really conclude anything about the mechanism how CAHLM6 may regulate immune function.

However, the authors tried many approaches to understand how CAHLM6 controls macrophage function. Among others they performed a transcriptomic analysis of CAHLM6^{-/-} vs wt mice. By comparison, they found some leads but did not follow up on that. They also tried the obvious experiment, to look at ATP release but did not see any differences between CAHLM6^{-/-} vs wt mice.

While the results of the present ms are interesting for the immune community, there is certainly room for improvement:

Major problems:

1. The synaptic localization is not clear to me from the pictures. I believe it should be possible, as shown in the cartoon (Fig. 7) to present immune synapse between macrophages and NK cells next to each other (on top of each other seems to make an analysis quite difficult) at different time points and explain a detailed quantification of the localization using these pictures. The authors need to invest more time into this and make the quantification more plausible.

2. I acknowledge that the authors tried several approaches to understand the mechanism of CAHLM6 action. However, since CAHLM channels are important for calcium homeostasis, it seems straightforward to analyze calcium signals in macrophages upon NK contact. This should not be difficult to do. In my opinion, the authors should attempt this and with their confocal microscope this should be possible.

Minor problems:

3. The authors state: "NK cells were easily distinguishable on top of BMDM based on cell size and morphology (Figure 5C). IS sites were imaged by serial Z-sections. At sites of NK cell contact with BMDM, we assessed polarization of actin and CALHM6 from BMDM towards the NK cells at 10-, 20- and 40-min post NK cells addition to IFN- γ and LPS-primed BMDM monolayer (Figure 5D). While at 0 min there was no polarization of CALHM6 towards NK cells in any of the contacts imaged, the percentage of IS that showed polarization of CALHM6 towards NK cells increases significantly with time, until 40 min when about 50% of all actin-ring positive contacts had polarized CALHM6-eGFP from BMDM towards the NK cell IS (Figure 5E)." Do I miss something here? Are there fluorescence pictures missing in Fig. 5C?

4. Depending on journal requirements, full WB should be shown in Figures.

Summary

Overall, I find the results regarding CAHLM6 function in macrophages quite interesting. However, I would only suggest publication of the ms if major points 1 and 2 are being addressed appropriately (regardless of the outcome of the calcium experiments).

Jelena Bezbradica, PhD
Associate Professor
Roosevelt Dr, Headington, Oxford OX3 7FY
Kennedy Institute of Rheumatology
University of Oxford
jelena.bezbradica@kennedy.ox.ac.uk

Karin Dumstrei
Senior Editor; Managing Editor,
EMBO Press

7 November 2022

Dear editor

Thank you for constructive feedback on our manuscript. We have incorporated requested changes and hope that manuscript is now improved and suitable for publication in EMBO. Below is our point-by-point response to review in blue.

Editor comment:

As you can see from the comments, the referees find the analysis interesting. However, they also all raise the issue that we would need some further mechanistic insight for consideration here. Referee #3 suggests to look at Calcium homeostasis as a possible mechanism. Should you be able to add more insight into how CALHM6 contributes to macrophage-NK cell communication then I would like to invite you to submit a revised version. We don't need the full mechanism but some is needed.

To dissect mechanism of CALHM6 action, we established a new collaboration with the laboratory of Prof J. Kevin Foskett, and a senior research investigator in his team, Dr Zhongming Ma. Prof Foskett runs an electrophysiology lab at University of Pennsylvania, with expertise in ion channels. His team was the first one to describe CALHM1 as an ATP channel important for cell-cell communication between taste-bud cells and sensory neurons, where ATP acts as a neurotransmitter (Ma et al PNAS 2012, Taruno et al, Nature 2013, Ma et al et Neuron 2018 etc). Collaboration with Foskett team allowed us to explore the mechanism of action of CALHM6 in immune system, and compare how conserved, or distinct, CALHM6 biology is from the biology of CALHM1. Summary of our findings is below and forms the basis of **revised Figure 5**, **revised Figure EV4** and **new Figure 6 and new Figure 7** of the revised manuscript.

Cryo-electron microscopy of recombinant CALHM6 revealed that it assembles into a large-pore channel (Drozdzyk *et al.*, 2020). However, the function of CALHM6 as an ion channel has not been previously demonstrated. Our biochemical results indicate that in BMDMs, CALHM6 monomers interact in a large oligomeric complex requiring disulphide bonds for its stability (**Figure 6A,B**), consistent with the structural observations of recombinant CALHM6. Using patch-clamp studies, we now also determined that when expressed in *Xenopus* oocytes, CALHM6 localized to the plasma membrane (**Figure 6C**) and formed a voltage-gated ion channel with pharmacological properties, including inhibition by Gd^{3+} and ruthenium red, similar to CALHM1 and CALHM1/3 channels when expressed in this system (**Figure 6F,G**). Mutation of a conserved acidic residue that inhibits CALHM1 channels also blocked the CALHM6-induced conductance (**Figure 6F,G**) without affecting its plasma membrane localization in oocytes (**Figure 6C**), indicating that the observed currents were mediated by CALHM6, via mechanism conserved with CALHM1. However, inhibition of CALHM6 conductance by extracellular divalent cations, i.e. Ca^{2+} , Ba^{2+} , is less effective than in CALHM1 and CALHM1/3 channels (**Figure 7A,B,C,D**), resulting in a constitutive open CALHM6-mediated plasma membrane permeability in oocytes, and striking cell toxicity (**Figure 6D**). To minimize toxicity of mammalian cells expressing CALHM6, either unknown regulatory mechanisms are present to ensure

channel closure, or it is not usually localized in the plasma membrane under resting conditions. The latter is consistent with our findings of dominant intracellular localization of CALHM6 in resting in BMDM (**Figure EV6A,B**) and N2a cells (**Figure 6E**), and with the lack of response of CALHM6-expressing N2a and HEK293T cells to a Ca^{2+} “add-back” protocol that activates membrane CALHM1 and CALHM1/3 channels (**Figure 7A,B,C,D**). Upon BMDM activation, CALHM6 was observed to translocate from intracellular compartments to the immunological synapse (**Figure 5D,F**). Thus, mammalian cells appear to restrict CALHM6 localization to intracellular compartments, from where, at least in macrophages, it gets recruited to the surface (**Figure 5D,F**) upon macrophage activation and synapse formation. We suggest that it is inserted into the membrane there where it provides a highly localized, and likely transient, permeability to metabolites that can permeate through its large pore to activate NK cells. In agreement, we find that CALHM6 upregulation facilitated the release of ATP (and possibly other signalling metabolites) only from activated but not resting macrophages (**Figure 7E**).

In conclusion, we provide novel insights into macrophage-NK cell immune synapse and into the role of CALHM6 as pathogen-induced, ATP-permeable, synaptic ion channel important for the timely activation of NK cells. Our results contribute to understanding of neurotransmitter-like signal exchange between immune cells that fine-tunes the timing of innate immune responses.

Referee #1:

In this MS the authors present a nice body of data to support that the membrane ion channel CALHM6 is an immune synaptic channel-forming protein that contributes to neurotransmitter-like signal exchange between myeloid and NK cells. The role of this protein seems to be restricted to the macrophage - NK cell synapse. The findings are both interesting and potentially important.

Major issues

1. The abstract says "they fail to control *L. monocytogenes* burden at the peak of infection". The results show CALHM6 is important for regulating the early innate control of *L. monocytogenes* infection. This is a bit misleading- it regulates the bacterial burden, but does not control it as the bacterial levels have dropped by D7 (which is interesting as what else kicks in to help regulate IFN γ production?). Presumably the IFN γ is also required for the development of clearing adaptive immunity against this pathogen as well? IL18 normally precedes the induction of IFN γ and the authors measure this later on, but some discussion a bit earlier in the MS might help to strengthen the arguments here. The IL18 data only plays a supporting role to IL1 in the MS at the moment, but it could be a reason why the differences in early/late NK cell IFN γ production is seen. Maybe the authors may need to reorder some of the data in the MS to address this? The data showing this may be more appropriate for earlier in the MS? Reword to say CALHM6 is important for regulating the early innate control of *L. monocytogenes* infection. Simple clarification of the various sources of IFN γ and its relationship to the different cell types involved in innate and adaptive immunity would be useful here to support the authors arguments.

We thank reviewer for these suggestions. We have changed the abstract, and text, to clarify that:

a) CALHM6 is important for regulating the early innate control of *L. monocytogenes*.

b) We showed that CALHM6 is not required to induce IL-18 secretion by *L. monocytogenes* infected macrophages, at least in vitro (**Figure 4E**). We tried to measure IL-18 in vivo after *L. monocytogenes* or Poly(I:C) in injection, but were unable to detect it in the serum, it was below the detection limit. However, we have repeated Poly(I:C) injection in vivo and measured the earliest innate immune responses to Poly(I:C) in WT or *Calhm6*^{-/-} mice, 1 h and 3 h post injection, i.e. before NK activation for IFN- γ production. We found that CALHM6 mice had no defect in detecting Poly(I:C) and secreting macrophage-derived cytokines and chemokines such as IFN- β , IL-12, IL-6, TNF, CCL22, and inflammasome-dependent IL-1 β (**Figure EV4K-R**). These results support the idea that *Calhm6*^{-/-} macrophages can detect and respond to Poly(I:C) normally, and that their early innate responses are not impaired.

c) In agreement with previous studies, we showed, and clarified better in revised text, that at earliest time point post *L. monocytogenes* and Poly(I:C) injection, NK1.1+ cells are the major source of IFN- γ and hence are the cells mostly affected by CALHM6 deficiency (**Figure 3C,H**).

2. "Cytokine response was then compensated, albeit with a delayed kinetics, to finally allow full bacterial clearance by day 7". This is a bit unclear: do the authors mean a hypothesis whereby NK cell IFN γ was reduced on D1, but IFN γ production from other cells compensated for this such that by D3 it was elevated? If so can the authors add to the text an explanation as to how and why this might happen.

NK cells are the earliest IFN- γ producers after *L. monocytogenes* infection and Poly(I:C) injection (**Figure 3C,H**). In *Calhm6*^{-/-} mice, this IFN- γ production in NK cells is nearly absent in the early hours and days of the immune response to Poly(I:C) and *L. monocytogenes*, but *Calhm6*^{-/-} mice could produce normal, albeit delayed IFN- γ response few days later, as detected by intracellular staining for IFN- γ in gated NK cells (**Figure 3A-J**). We have reordered figure panels to make the delayed kinetic in NK cell activation clear. We also discussed possible explanation, based on our data, for this delay in NK cell activation. The most efficient and fastest way to activate NK cells *in vivo* is via direct contact between activated myeloid cells and NK cells through the formation of an IS (Newman & Riley, 2007). Because CALHM6 acts as an ion channel in macrophage-NK cell synapse, we conclude that there is a deficit in the early contact-dependent communication between myeloid cells and NK cells in *Calhm6*^{-/-} mice. But *Calhm6*^{-/-} macrophages can detect bacteria and Poly(I:C) normally *in vivo* and can secrete soluble cytokines essential for NK cells activation (**Figure EV4**). We also showed that NK cells can respond to these cytokines (**Figure 4C**), likely explaining why NK cells eventually do become activated, albeit with a delayed kinetics in *Calhm6*^{-/-} mice.

3. "so the early defect in NK cell derived IFN-g would explain uncontrolled proliferation of the bacteria" better here to say enhanced proliferation of the bacteria as there is no evidence to show the bacterial growth is uncontrolled: this would require a number of different experiments in different KO mouse strains which is beyond the scope of this MS

Thank you, this is correct. Text has been edited to make it clear that *L. monocytogenes* proliferation is enhanced due to the delay in innate immune responses but is not fully uncontrolled, as mice manage to restore control over infection by day 7.

4. The images in Fig 5 D-F are too small and it is difficult to see what is happening. Can the authors get better images supporting CALHM6 localisation to the NK immune synapse?

Thank you, we have now provided new image (**Figure 5D**) to replace the old one that showed just one confocal slice through the cell and was focused on the plane of macrophage-NK cell interaction. In revised figure, immune synapse sites were imaged by serial Z-sections; and an entire collapsed Z-stack is shown in Figure 5D. It demonstrates CALHM6 recruitment from the intracellular compartment to the membrane synapse where macrophages are establishing communication with NK cells.

5. Do the authors have any hypothesis/explanation as to why CALHM6 would only be found in the macrophage - NK cell synapse? One might have expected a IS regulatory protein like this to be present in all the antigen presenting cells that form ISs? Does another CALHM perhaps perform similar functions in DCs?

Reviewer is correct, our focus on macrophages in this study, does not exclude the possibility that CALHM6 is important in DCs in other conditions (e.g. upon detection of tumours, where its expression may be induced at higher levels (*Chiba et al, 2014*)). This has been now clarified in text. We found that upon detection of pathogens, endogenous CALHM6 expression is highest in macrophages (**Figure 1A,B**), so we used macrophages as the most relevant cell for our study. This was supported by our finding that deletion of CALHM6 in dendritic cells alone (CD11cCre) was not sufficient to recapitulate phenotype of CALHM6 KO mice (**Figure EV5C,D**) during infection.

Minor issues

Title: What does synaptic channel mean? Consider changing the title to immune synaptic channel protein for clarity

Title has been changed as suggested.

Abstract

"the timing of innate immune response" to: the timing of innate immune responses
Abstract has been changed as suggested.

Referee #2:

The authors investigate the role of the CALMH6 protein (also known as INAM), a voltage-gated oligomeric ATP channel, for NK cell activation and NK-myeloid cell crosstalk. In extension of previous data, they show that CALMH6 is the only molecule of this family which is inducibly expressed by PAMP-activated myeloid cells. A newly generated *Calmh6*^{fl/fl} mouse line shows that control of *Listeria monocytogenes* is impaired at day 3 but bacterial clearance by day 7 is not affected. While *Calmh6*^{-/-} NK cells were impaired in their propensity to produce IFN γ early after stimulation (polyIC or *L. monocytogenes*) they produced normal levels of IFN γ at later timepoints. Cell type-specific deletion of *Calmh6* in dendritic cells did not lead to any differences in NK cell granzyme B and IFN γ expression. Genome-wide transcriptional analysis of bone marrow-derived macrophages of *Calmh6*^{-/-} and control mice revealed only few differentially expressed genes, many of which were part of synaptic pathways. In particular genes associated with vesicle trafficking and fusion, or formation of synaptic contacts were reduced in *Calmh6*^{-/-} macrophages. Biochemical and CryoEM studies show that CALMH6 forms an oligomer and may act as a channel but the signals opening the CALMH6 channel and thereby affecting NK cell function remain unknown.

General concerns

The paper addresses an interesting topic, how CALMH6 enhances NK cell activation possibly by amplifying NK cell-macrophage crosstalk. The presented data is of good quality but, overall, new insights are limiting. Much of the data confirms previous work in another disease model (refs. 4, 5). A newly generated tissue-specific allele of the *Calmh6* gene is a valuable tool but its potential is not fully explored. **The most interesting experiments are in Figures 5 and 6 but mainly negative data are presented. From the data presented, it remains unclear how CALMH6 amplifies NK cell function.** More mechanistic insights would be expected from a paper in EMBO J.

Thank you for pointing out the most novel area that needed improvement. We have now added revised Figure 5, and new Figure 6 and new Figure 7 to include mechanistic data on CALMH6 function, and to compare it with neuronal CALHM1. Data are described in detail in response to editor above and summarised in our point-by-point response below.

The following specific concerns are raised.

1. The paper does not provide a mechanistic understanding of how CALMH6 contributes to NK cell activation. Without such data the significance is rather limiting. Further exploration of CALMH6 in the immunological synapse or its function as a channel would be needed.

We have now provided new data to show that CALHM6, like CALHM1, also forms a, ATP-permeable membrane ion channel. Opening of CALHM6 is regulated by changes in membrane voltage and by a conserved negatively charged residue E119R in the extracellular loop. Conserved acidic residue D121 also controls opening of CALHM1 family (Ma, et al. PNAS, 2012). But unlike CALHM1, CALHM6 is not regulated by extracellular divalent cations so CALHM6 is open and therefore toxic to cells if it is constitutively present in the plasma membranes like in *Xenopus* oocytes. So, we found that mammalian cells have found a way to keep CALHM6 location restricted to the intracellular compartment in resting state. From there, CALHM6 is recruited, on-demand, to immune synapses upon macrophage activation. Its function is thus regulated by macrophage activation, by trafficking and likely by additional NK cell derived signals that regulate its opening within immune synapse.

2. Previous data suggested that NK cells can also express CALMH6? How does NK cell function look in mice with a deletion of Calmh6 gene in NK cells (Ncr1-Cre).

We found that expression of CALHM6 in NK cells was barely detectable when compared to macrophages (not shown). We do however agree that CALHM6 may play function in both macrophages and NK cells. To better understand the cell-type specific role of CALHM6 (macrophages, DCs, NK cells), we tried to optimise a mixed *in vitro* co-culture system based on the one established by Kasamatsu and colleagues (Kasamatsu *et al.*, 2014). We co-cultured either BMDCs, BMDM, or freshly isolated F4/80⁺ cells from WT or *Calhm6*^{-/-} spleens with either magnetically- or FACS-sorted WT or *Calhm6*^{-/-} NK cells, in the presence or absence of Poly(I:C). However, none of these co-culture conditions were reliably recapitulated the delayed NK cell activation phenotype seen *in vivo* (Figure EV5E to G). A similar result was found using *Rag*^{-/-}*Calhm6*^{-/-} mice, which do not develop adaptive immune cells, therefore avoiding any contribution of T cells to IFN- γ secretion (Figure EV5H). We hypothesized that any defects in contact-dependent activation of NK cells in *Calhm6*^{-/-} co-cultures must happen quite early *in vitro*, and that accumulation of NK-activating soluble cytokines (e.g. IL-12, IL-18) over time in a small volume of the 96-well co-culture dish quickly compensates for the initial delay in activation, making the experiment subject to significant variability. Thus, we agree with reviewer that the only reliable way to address CALHM6 biology in NK cells is to make Ncr1-Cre;*Calhm6*^{fl/fl} mice. We are now generating this strain, as well as LysMCre *Calhm6*^{fl/fl} fl. They were unfortunately not ready on time for this revision, so will be characterised in detail in a follow up study.

3. The authors present data from mice lacking CALMH6 in dendritic cells (using CD11c-Cre) which did not change NK cell function. This should be further explored by using deletion of CALMH6 in macrophages, pDC, NK cells or all immune cells (Vav-Cre).

We agree, we have addressed this issue in response to reviewer 2 question 2 and reviewer 1 question 5. Previously, CALHM6 expression was shown to be restricted to CD45⁺ cells, and ImmGen data base shows highest CALHM6 expression on myeloid cells in mice and human. Our future work will focus on biology of CALHM6 in CD11cCre, Ncr-1Cre and LysMCre *Calhm6* fl/fl mice. They were unfortunately not ready on time for this revision, so will be characterised in detail in a follow up study.

4. The establishment of a co-culture system of NK cells and myeloid (or other accessory cells) that would allow to assess how CALMH6 enhances NK cell function would be extremely helpful to gain additional mechanistic insights.

We have addressed this issue in response to reviewer 2 question 2 and 3, and reviewer 1 question 5.

5. The authors claim that the production of type I IFNs is not altered in *Calmh6*^{-/-} mice. However, this is based on the assessment of type I IFN response genes (Figure S4). Direct measurements of serum type I IFNs should be provided.

As suggested, we have repeated Poly(I:C) injection and measured the earliest innate immune responses to Poly(I:C) in WT or *Calhm6*^{-/-} mice, 1 h and 3 h post injection, i.e. before NK activation for IFN- γ production. We found that CALHM6 mice had no defect in detecting Poly(I:C) and secreting macrophage-derived cytokines and chemokines such as IFN- β , IL-12, IL-6, TNF, CCL22, and inflammasome-dependent IL-1 β (Figure EV4K-R). These results support idea that suggesting that *Calhm6*^{-/-} macrophages can detect and respond to Poly(I:C) normally, and that their early innate responses are not impaired.

Referee #3:

CALHM channels are ion channels with some features reminiscent of VRAC, pannexins, connexins. They are believed to conduct cations (with poor discrimination), anions and ATP. Potential functions of CAHLM6 in the immune systems are not well understood. The present ms by Danielli et al analyzes CALHM6 (Calcium homeostasis modulator ion channels) function in macrophages.

Pros

The authors convincingly show that CAHLM6 is the only CAHLM channel expressed in murine macrophages upon pathogen-dependent stimulation. They clearly show that CAHLM6 expression is upregulated by

proinflammatory signals and downregulated by anti-inflammatory ones. The authors furthermore report that CAHLM6^{-/-} animals control infection by the bacteria *Listeria monocytogenes* not as well as wt mice with IFN-gamma production by NK cells being delayed/decreased. Immune development is, however, normal. Finally, the authors try to show that CAHLM6 is segregated to the immune synapse between macrophages and NK cells, which kind of links the macrophage CAHLM6 expression with the NK problems in maintaining high INF-gamma levels. With respect to immune synapse segregation of CAHLM6 I see some problems (see below).

Cons

The authors cannot really conclude anything about the mechanism how CAHLM6 may regulate immune function. However, the authors tried many approaches to understand how CAHLM6 controls macrophage function. Among others they performed a transcriptomic analysis of CAHLM6^{-/-} vs wt mice. By comparison, they found some leads but did not follow up on that. They also tried the obvious experiment, to look at ATP release but did not see any differences between CAHLM6^{-/-} vs wt mice.

We agree, mechanism of CALHM6 function was the weakest point of the original manuscript and hence was the focus of our revisions. To dissect mechanism of CALHM6 action, we established a new collaboration with the laboratory of Prof J. Kevin Foskett, and senior research investigator in his team, Dr Zhongming Ma. Prof Foskett runs an electrophysiology lab at University of Pennsylvania, with expertise in ion channels. His team was the first one to describe CALHM1 as an ATP channel important for cell-cell communication between taste-bud cells and sensory neurons, where ATP acts as a neurotransmitter (Ma et al PNAS 2012, Taruno et al, Nature 2013, Ma et al et Neuron 2018 etc). Collaboration with Foskett team allowed us to explore the mechanism of action of CALHM6 in immune system, and compare how conserved, or distinct, CALHM6 biology is from the biology of CALHM1. Summary of our findings is below and forms the basis of revised Figure 5, revised Figure EV4 and new Figure 6 and new Figure 7 of the revised manuscript. Results are summarised in response to editor above and Reviewer 2 question 1.

While the results of the present ms are interesting for the immune community, there is certainly room for improvement:

Major problems:

1. The synaptic localization is not clear to me from the pictures. I believe it should be possible, as shown in the cartoon (Fig. 7) to present immune synapse between macrophages and NK cells next to each other (on top of each other seems to make an analysis quite difficult) at different time points and explain a detailed quantification of the localization using these pictures. The authors need to invest more time into this and make the quantification more plausible.

Thank you, we have now provided new image (Figure 5D) to replace the old one that showed just one confocal slice through the cell and was focused on the plane of macrophage-NK cell interaction. In the revised figure, immune synapse sites were imaged by serial Z-sections; and an entire collapsed Z-stack is shown in Figure 5D. It demonstrates CALHM6 recruitment from the intracellular compartment to the membrane synapse where macrophages are establishing communication with NK cells.

2. I acknowledge that the authors tried several approaches to understand the mechanism of CAHLM6 action. However, since CAHLM channels are important for calcium homeostasis, it seems straightforward to analyze calcium signals in macrophages upon NK contact. This should not be difficult to do. In my opinion, the authors should attempt this and with their confocal microscope this should be possible. We have analysed ion fluxes and Ca²⁺ fluxes in *Xenopus* oocytes expressing CALHM1, CALHM6 or channel dead E119 mutant of CALHM6. Using patch-clamp studies, we now determined that when expressed in *Xenopus* oocytes, CALHM6 localized to the plasma membrane (Figure 6C) and formed a voltage-gated ion channel with pharmacological properties, including inhibition by Gd³⁺ and ruthenium red, similar to CALHM1 and CALHM1/3 channels when expressed in this system (Figure 6F,G). Mutation of a conserved acidic residue that inhibits CALHM1 channels also blocked the CALHM6-induced conductance (Figure 6F,G) without affecting its plasma membrane localization in oocytes (Figure 6C), indicating that the observed currents were mediated by CALHM6, via mechanism conserved with CALHM1. However, inhibition of

CALHM6 conductance by extracellular divalent cations, i.e. Ca^{2+} , Ba^{2+} , is less effective than in CALHM1 and CALHM1/3 channels (**Figure 7A,B,C,D**), resulting in a constitutive open CALHM6-mediated plasma membrane permeability in oocytes, leading to striking cell toxicity (**Figure 6D**). To minimize toxicity of mammalian cells expressing CALHM6, either unknown regulatory mechanisms are present to ensure channel closure, or it is not usually localized in the plasma membrane under resting conditions. The latter is consistent with our findings of dominant intracellular localization of CALHM6 in resting in BMDM (**Figure EV6A,B**) and N2a cells (**Figure 6E**), and with the lack of response of CALHM6-expressing N2a and HEK293T cells to a Ca^{2+} "add-back" protocol that activates membrane CALHM1 and CALHM1/3 channels (**Figure 7A,B,C,D**). However, during BMDM activation, CALHM6 was observed to translocate from intracellular compartments to the immunological synapse (**Figure 5D,F**). We suggest that it is inserted into the membrane there where it provides a highly localized, and likely transient, permeability to metabolites that can permeate through its large pore to activate NK cells. In agreement, we find that CALHM6 upregulation facilitated the release of ATP (and possibly other signalling metabolites) only from activated but not resting macrophages (**Figure 7E**).

Our results are most consistent with a model in which CALHM6 mediates ATP release from activated macrophages that requires several steps: first, CALHM6 expression is upregulated upon macrophage activation, second, CALHM6 is recruitment to the plasma membrane at the immunological synapse, and third, CALHM6 channel opening is enhanced, possibly by local signals in the IS. Such stepwise control may be necessary in mammalian cells to prevent toxicity observed in CALHM6-expressing oocytes.

Minor problems:

3. The authors state: "NK cells were easily distinguishable on top of BMDM based on cell size and morphology (Figure 5C). IS sites were imaged by serial Z-sections. At sites of NK cell contact with BMDM, we assessed polarization of actin and CALHM6 from BMDM towards the NK cells at 10-, 20- and 40-min post NK cells addition to IFN-g and LPS-primed BMDM monolayer (Figure 5D). While at 0 min there was no polarization of CALHM6 towards NK cells in any of the contacts imaged, the percentage of IS that showed polarization of CALHM6 towards NK cells increases significantly with time, until 40 min when about 50% of all actin-ring positive contacts had polarized CALHM6-eGFP from BMDM towards the NK cell IS (Figure 5E)." Do I miss something here? Are there fluorescence pictures missing in Fig. 5C?

Text has been edited to correct the typo and incorrect calling of figures. Also, we have now provided new image (**Figure 5D**) to replace the old one that showed just one confocal slice through the cell and was focused on the plane of macrophage-NK cell interaction. In the revised figure, immune synapse sites were imaged by serial Z-sections; and an entire collapsed Z-stack is shown in Figure 5D. It demonstrates CALHM6 recruitment from the intracellular compartment to the membrane synapse where macrophages are establishing communication with NK cells.

4. Depending on journal requirements, full WB should be shown in Figures. These have been provided now as source data.

Summary

Overall, I find the results regarding CALHM6 function in macrophages quite interesting. However, I would only suggest publication of the ms if major points 1 and 2 are being addressed appropriately (regardless of the outcome of the calcium experiments). Thank you for the support and constructive criticism which have focused our attention on mechanism and hopefully improved the final manuscript.

Dear Jelena,

Thank you for submitting your revised manuscript to The EMBO Journal. Your study has now been re-reviewed by referees #2 and 3. As you can see from the comments below, both referees appreciate the introduced changes and support publication here. I am therefore very pleased to accept the MS for publication here. Before sending you the formal accept letter there are just a few editorial points to resolve:

- The funding information John Fell Oxford University Press Research Fund (0010732) is missing in the online submission system.
- Please relabel "Data and materials availability" to "data availability" and list the GEO accession for the RNA sequencing data in this section. Also remove "The research materials supporting this publication can be accessed by contacting corresponding author jelena.bezbradica@kennedy.ox.ac.uk and foskett@pennmedicine.upenn.edu" from this section.
- COI needs renaming to "DISCLOSURE AND COMPETING INTERESTS STATEMENT"
- Please remove the Authors Contributions from the manuscript. The 'Author Contributions' section is replaced by the CRediT contributor roles taxonomy to specify the contributions of each author in the journal submission system. Please use the free text box in the 'author information' section of the manuscript submission system to provide more detailed descriptions (e.g., 'X provided intracellular Ca⁺⁺ measurements in fig Y')
- Please check figure callouts for Figure 6F, 6G, 8A and 8B
- Supp Table 1-7 should be renamed to Dataset EV1-EV7 with a legend in a separate tab of each Excel file. Please also correct call out.
- Our publisher has also done their pre-publication check on your manuscript. When you log into the manuscript submission system you will see the file "Data Edited Manuscript file". Please take a look at the word file and the comments regarding the figure legends and respond to the issues.
- We don't encourage showing statistic when n=2 (Fig 1H, Fig 7D, EV6G, EV7C)
- We include a synopsis of the paper that is visible on the html file (see <http://emboj.embopress.org/>). Can you provide me with a general summary statement and 3-5 bullet points that capture the key findings of the paper?
- I also need a summary figure for the synopsis. The size should be 550 wide by [200-400] high (pixels).
- you can only have 5 EV figures - Is it possible to combine a few of them? Also, EV figures need to be uploaded as separate files, and legends included in the MS file
- Author emails bounced for Usha Paudel and Benjamin Demarco.

Please also submit a point-by-point response when you send in your revised version.

Congratulations on a nice study

Best Karin

Karin Dumstrei, PhD
Senior Editor
The EMBO Journal

Guide For Authors: <https://www.embopress.org/page/journal/14602075/authorguide>

Use the link below to submit your revision:

Referee #2:

This is a revised version of the manuscript that has undergone significant changes. While the full mechanism of how CALMH6 contributes to NK-macrophage interactions remains unclear, additional data (patch clamp studies mainly in *Xenopus* oocytes) now clarify some of the electrophysiological features of the CALMH6 channel. In addition, interesting data about the subcellular localization of CALMH6 is provided. I find the manuscript improved, although the exact mechanism by which CALMH6 signaling is involved in macrophage-NK cell crosstalk remains unclear.

Referee #3:

My major point 1.

Original review: The synaptic localization is not clear to me from the pictures.

Revised version: new images are provided. These are convincing (Fig. 5D). The analysis (Fig. 5E) is identical to the original version. I checked to original version again and I do think that the analysis is solid enough.

My major point 2.

Original review: I acknowledge that the authors tried several approaches to understand the mechanism of CALMH6 action. However, since CALMH channels are important for calcium homeostasis, it seems straightforward to analyze calcium signals in macrophages upon NK contact. This should not be difficult to do. In my opinion, the authors should attempt this and with their confocal microscope this should be possible.

Revised version: The authors have involved an expert ion channel lab. They provide now new and important data and characterized CALMH6 currents in oocytes. These data are very solid. They could also show that ATP release was facilitated upon CALMH6 upregulation. Together these data allow a better mechanistic picture of CALMH6 function at the IS. To allow partial mechanistic insight is the main point of the editor's original comments and this is in my opinion fulfilled with these new data (Fig. 6, 7). Even if the role of calcium is still unclear, I acknowledge the great effort by the authors (and in particular also by the new authors!) to unmask at least parts of the mechanism.

I was not able to open the source data to have a look at the Western Blots. I assume that full blots are shown.

Summary: The authors have responded adequately to my major and minor points and in my opinion they have also addressed to important points by the editor and the other reviewers. I thus suggest acceptance of the ms.

Jelena Bezbradica, PhD
Associate Professor
Roosevelt Dr, Headington, Oxford OX3 7FY
Kennedy Institute of Rheumatology
University of Oxford
jelena.bezbradica@kennedy.ox.ac.uk

Karin Dumstrei
Senior Editor; Managing Editor,
EMBO Press

19 January 2023

Dear editor

Thank you for constructive and positive feedback on our manuscript. We have incorporated requested final changes and hope that manuscript is now suitable for publication in EMBO. Below is our point-by-point response to editor review in blue.

- The funding information John Fell Oxford University Press Research Fund (0010732) is missing in the online submission system.

We have corrected funding information in the submission system

- Please relabel "Data and materials availability" to "data availability" and list the GEO accession for the RNA sequencing data in this section. Also remove "The research materials supporting this publication can be accessed by contacting corresponding

author jelena.bezbradica@kennedy.ox.ac.uk and foskett@pennmedicine.upenn.edu" from this section. We have made the requested change and provided GEO data accession for the RNA sequencing. Data will go live on line on February 1st.

- COI needs renaming to "DISCLOSURE AND COMPETING INTERESTS STATEMENT"

We have made the requested change.

- Please remove the Authors Contributions from the manuscript. The 'Author Contributions' section is replaced by the CRediT contributor roles taxonomy to specify the contributions of each author in the journal submission system. Please use the free text box in the 'author information' section of the manuscript submission system to provide more detailed descriptions (e.g., 'X provided intracellular Ca⁺⁺ measurements in fig Y')

We have moved information about contribution to the 'author information' section.

- Please check figure callouts for Figure 6F, 6G, 8A and 8B

This has been checked. Figure 8 is now deleted and provided as synopsis figure

- Supp Table 1-7 should be renamed to Dataset EV1-EV7 with a legend in a separate tab of each Excel file. Please also correct call out.

We have merged all tables into a single file, that is submitted under name Appendix Dataset EV1

- Our publisher has also done their pre-publication check on your manuscript. When you log into the manuscript submission system you will see the file "Data Edited Manuscript file". Please take a look at the word file and the comments regarding the figure legends and respond to the issues.
We have made all requested edits, they are marked in red in all figure legends.

- We don't encourage showing statistic when n=2 (Fig 1H, Fig 7D, EV6G, EV7C)
These statistics have been removed.

- We include a synopsis of the paper that is visible on the html file (see <http://emboj.embopress.org/>). Can you provide me with a general summary statement and 3-5 bullet points that capture the key findings of the paper?

We have included summary statement and 5 bullet points to go along synopsis figure.

- I also need a summary figure for the synopsis. The size should be 550 wide by [200-400] high (pixels).
We have provided Synopsis figure. It has part A and part B. If this is too large, please use only part A.

- you can only have 5 EV figures - Is it possible to combine a few of them? Also, EV figures need to be uploaded as separate files, and legends included in the MS file

We have merged EV figures to total of 5 figures.

- Author emails bounced for Usha Paudel and Benjamin Demarco.

We have updated email addresses in the system.

Thank you for the support and constructive criticism which have focused our attention on mechanism and improved the final manuscript.

Best wishes

Jelena Bezbradica

Dear Jelena,

Thank you for submitting your revised manuscript to The EMBO Journal. I have now carefully looked at everything and all looks good.

I am therefore very pleased to accept the manuscript.

Congratulations on a nice study!

best Karin

Karin Dumstrei, PhD
Senior Editor
The EMBO Journal

Please note that it is EMBO Journal policy for the transcript of the editorial process (containing referee reports and your response letter) to be published as an online supplement to each paper. If you do NOT want this, you will need to inform the Editorial Office via email immediately. More information is available here:

<https://www.embopress.org/page/journal/14602075/authorguide#transparentprocess>

Your manuscript will be processed for publication in the journal by EMBO Press. Manuscripts in the PDF and electronic editions of The EMBO Journal will be copy edited, and you will be provided with page proofs prior to publication. Please note that supplementary information is not included in the proofs.

You will be contacted by Wiley Author Services to complete licensing and payment information. The required 'Page Charges Authorization Form' is available here: https://www.embopress.org/pb-assets/embo-site/tej_apc.pdf - please download and complete the form and return to embopressproduction@wiley.com

EMBO Press participates in many Publish and Read agreements that allow authors to publish Open Access with reduced/no publication charges. Check your eligibility: <https://authorservices.wiley.com/author-resources/Journal-Authors/open-access/affiliation-policies-payments/index.html>

Should you be planning a Press Release on your article, please get in contact with embojournal@wiley.com as early as possible, in order to coordinate publication and release dates.

If you have any questions, please do not hesitate to call or email the Editorial Office. Thank you for your contribution to The EMBO Journal.
